# Developing an Evaluation System Suitable for Coastal Rural Houses’ Characteristic Style and Its Inspiration for Rural Revitalization: Case Study of Rongcheng in Shandong Province

**DOI:** 10.3390/ijerph20043010

**Published:** 2023-02-09

**Authors:** Qinglei Zhao, Guanghui Jiang, Tao Zhou, Wenqiu Ma, Yuting Yang

**Affiliations:** 1College of Geography and Tourism, Qufu Normal University, Rizhao 276826, China; 2State Key Laboratory of Earth Surface Process and Resource Ecology, School of Natural Resources, Faculty of Geographical Science, Beijing Normal University, Beijing 100875, China; 3College of Engineering, China Agricultural University, Beijing 100083, China; 4Department of Geography, Ghent University, 9000 Ghent, Belgium

**Keywords:** coastal rural houses, traditional village, characteristic style, evaluation, regionalization, Rongcheng

## Abstract

The characteristic style of rural houses is an important manifestation of the historical and cultural values of rural areas and is the key focus of the implementation of the strategy for the construction of beautiful China and the revitalization of rural areas. Taking 17 villages in the Rongcheng of Shandong as an example, this article integrated multidimensional data, including geospatial data, survey data and socio-economic data, and constructed a suitable index system to evaluate the characteristic style of coastal rural houses in 2018 and put forward the characteristic style regionalization of coastal rural houses. The results show that the characteristic style of coastal rural houses can be measured by the overall village environment, coastal architectural value and traditional folk culture, among which the coastal architectural value is the most critical. Two villages scored over 60 points in the comprehensive evaluation, namely the Dongchu Island village and Dazhuang Xujia Community. Different dominant characteristic styles of rural houses were identified according to single-factor evaluation. Based on the evaluation results and factors such as location, nature, social economy and the status quo of protection and development management, characteristic styles of rural houses in the research area can be divided into four continuous areas: historical and cultural characteristics, folk customs and industrial development characteristics, natural scenery characteristics and folk customs characteristics. Combined with regional positioning and development planning, the construction direction of different regional types was defined, and then the protection and improvement measures of rural residential features were put forward. This study not only provides a certain basis for the evaluation, construction and protection of the characteristic features of coastal rural dwellings in Rongcheng City but also provides guidance for the implementation of rural construction planning.

## 1. Introduction

Landscape style is the overall appearance of the environment that can reflect historical and cultural characteristics [1]. Rural development is an important basis for China’s economic development [2], and the protection and development of the residential features of traditional villages is an important part of rural development [3]. The characteristics of rural dwellings are the village environment and the customs and culture embodied in the natural environment around the village, the buildings in the village and the intangible cultural heritage [4]. The characteristic style of rural dwellings is an organic synthesis, reflecting the historical and cultural value of the village.

The quantitative evaluation of rural houses’ characteristics is of great significance to the investigation and protection of the current situation of characteristic villages. It can deepen the public’s understanding of the characteristics of rural dwellings and provide reference for the subsequent zoning research and the formulation of protection policies [5,6]. Most current studies choose historical and cultural villages, ethnic minority villages or urban waterfront spaces as the study area and analyze and evaluate its characteristics [7,8,9,10,11,12,13]. There are few quantitative evaluations of the characteristic style of coastal rural houses. The style investigation of rural houses’ characteristics is the basis of evaluation, and the authenticity and reliability of survey data are the premise to ensure the accuracy of evaluation [14,15,16].

At present, the commonly used research methods are mainly literature collection and collation, field investigation, comprehensive analysis of data classification and quantitative analysis of assessment indicators [17,18]. These methods are based on statistical theory, showing attribute data, and can show less spatial information. The style regionalization of rural houses’ characteristics is based on its evaluation, combined with the differentiated development conditions and directions of different villages, and it provides spatial guidance for the implementation of policies so as to promote the formation of rural style patterns with significant style zoning and clear style features [19,20,21]. Some scholars constructed an evaluation index system, identified different rural types and successively studied the classification of rural functions, the types of rural space and the level of rural economic diversification [22,23]. Other scholars identified different regional rural development types and evaluated rurality [24]. At present, the regional distribution of rural type division and rurality research is extensive, and the research methods are diverse.

Different rural types have different styles, with different rural house characteristics. Based on this, the academic community has carried out research on the classification of rural residential areas and the evaluation of urban features and the status quo of coastal characteristic seagrass houses and proposed protection-planning measures, the direction of protection and development of traditional village features, the framework and technical route of historical and cultural village protection planning based on style features and the reference significance of traditional residential research to rural settlements in various disciplines [25,26]. In addition, with the continuous impact of urban development on rural areas, the relevant research is concerned with how traditional village styles can maintain their own rural characteristics and avoid forming homogenized villages, and it suggests that the homogenization situation of urban and rural has become a major challenge [27,28,29,30,31].

Following the concept of ecological civilization, the concept of “beautiful China” has gradually formed. Agriculture is an important foundation of our country, and rural beauty will have the beauty of China. At present, the commonly used methods of style surveys display attribute data, and few can display spatial information. Most of the studies on rural style only aimed at famous historical and cultural villages, ancient villages and ethnic minority villages, and there is a lack of detailed study on the quantitative evaluation of the characteristic style of coastal rural dwellings. The coastal rural villages in Rongcheng, represented by seaweed houses, have become an important feature of this city, and its research has received increasing attention. The rural residences in Rongcheng, Shandong, are very distinctive, and the unique seagrass houses constitute the distinctive style of the area. Strengthening the style evaluation and zoning of coastal rural houses’ characteristics in the Rongcheng area can preserve it more completely and truly. Protecting the characteristic style of coastal rural houses that have not yet been valued can further protect China’s historical and cultural heritage for the subsequent formulation of protection policies and protection methods.

The purpose of this paper is to study the characteristic style of rural houses in a coastal city. This paper attempts to construct an index system suitable for the characteristic style of rural dwellings, takes Rongcheng City, Shandong, as the research area, evaluates the rural house characteristics of 17 villages integrating multidimensional data, including statistical data and survey-measured data in 2018, identifies the style types of village dwellings, reveals the characteristics and development direction of each region, provides policy recommendations for subsequent targeted protection and development and enriches the research in the style field of rural house characteristics. Through quantitative evaluation and zoning research, development conditions and directions are clarified, and corresponding measures are set to protect and enhance the value of the rural characteristic style, which is connected to the rural revitalization strategy and complements the construction of beautiful China [32,33,34,35,36].

The rest of this paper is organized as follows (Figure 1). Section 2 puts forward the concepts and theoretical underpinning. Section 3 summarizes the sources of data and the selection of a typical study area and introduces the methodology used in our study. Section 4 measures the characteristic style of rural houses in Rongcheng, Shandong, in 2018. Section 5 discusses the contribution to research, limitations and future work and puts forward policy recommendations for the protection and development of villages with different styles. Section 6 draws the conclusions.

## 2. Theoretical Underpinning

### 2.1. Rural Houses’ Characteristic Style

In general, “style” refers to the substantial external shape of the tangible material entity and the spiritual appearance displayed. It can be understood in two parts: the first part emphasizes the “inner”, which is condensed to the characteristics of the social and cultural orientation of this region, including the characteristics of the historical and cultural background, local conditions and people’s conditions, customs and culture; the other part emphasizes the “external”, which is the embodiment of the overall environmental characteristics and landscape overview and is the face, form and appearance of the area, including mountains, the sea, vegetation, climate and other natural environmental characteristics. Both supplement each other. The characteristic style of rural houses is the village environment and custom culture reflected by the natural environment around the village, the buildings in the village and the intangible cultural heritage, which are an organic complex [37].

The purpose of the enhancement of the characteristic architectural style is to build a good overall architectural image and a humanistic architectural image and to guide the factors in architectural features correctly so as to guarantee the harmonious beauty view of architectural features and directly reflect the value of characteristic culture. The improvement of the characteristic architectural style can realize the harmonious coexistence between folk houses and nature and integrate closely with the rich characteristic culture. The improved characteristic architectural style enables people to directly experience and feel the further improvement of space, environment and quality (Figure 2).

### 2.2. Index System of Rural Houses’ Characteristic Style

The characteristic style of rural houses is formed under a certain geographical space, combined with characteristic historical and cultural factors. The “Evaluation index system of Chinese historical and cultural famous towns (famous villages)” and the “Evaluation and Recognition Index System of Traditional Villages (Trial)” issued by the Ministry of Housing and Urban–Rural Development of the People’s Republic of China were constructed based on material and intangible cultural heritage to evaluate the characteristics of villages (towns).

According to the authors’ previous investigation and research, the evaluation index factor of coastal rural features can be adjusted appropriately on the basis of it. The overall village environment is a comprehensive expression of the rural location, village pattern, streets and courtyards and reflects the pattern and form of the traditional village. The characteristic style value of a rural house is the sum of tangible and intangible values of the whole village, which is mainly reflected by building value and traditional folk cultural value. Evaluation and regionalization provide spatial guidance for the implementation of differentiated management policies, so it is necessary to consider the protection, development and management of rural house characteristics so as to make it more targeted and practical. In order to evaluate the characteristic style of rural houses in the coastal area of Rongcheng, a suitable evaluation index system should be constructed according to the characteristics of the villages. The main style of coastal residential buildings in Rongcheng is that there are a large number of seagrass houses. The distribution and preservation status of seagrass houses in different villages are different, so each has its own characteristics.

To sum up, based on the research of relevant scholars and the advice of experts [11,12,13,14,15,16], this study takes into account the overall village environment and the characteristics of coastal buildings in the quantitative evaluation of coastal rural house features and combines them with the characteristics of traditional folk culture to build a suitable evaluation index system. The regionalization of the characteristic style of rural houses first considers the evaluation results, then combines the status quo of protection, development and management, analyzes the socio-economic conditions and location conditions of villages and finally considers the overall environment and natural conditions of villages so as to comprehensively identify different villages and type areas (Figure 2).

The index weight determined by a subjective weighting method can reflect the intention of researchers, but human factors limit the application of actual data. However, the weights determined by an objective weighting method are closely related to actual data, although they are susceptible to extreme values [36,38]. Therefore, this study uses a comprehensive weight determination method combining subjective and objective weighting methods and chooses a classical subjective weighting method, the analytic hierarchy process (AHP), and an objective weighting method reflecting actual data, the entropy method. When combined, the two methods complement each other and reduce the limitations of a single method. The index system constructed in this study consists of four layers. For the comprehensive evaluation layer (B) and the factor evaluation layer (C), the analytic hierarchy process is used to calculate the weight of the target layer (A). For the subfactor evaluation layer (D), the entropy method is used to calculate the relative weight. The weight of each factor calculated by the two methods is multiplied to obtain the weight of the comprehensive subjective and objective weighting methods.

## 3. Methods, Region and Data Processing

### 3.1. Study Area

Rongcheng is located in the easternmost part of Shandong Province and belongs to Weihai. The latitude and longitude ranges are 122°08′–122°42′ E and 36°45′–37°27′ N, respectively. Rongcheng City is surrounded by sea on three sides, the terrain is high in the north and low in the south, and the average altitude is 25 m. It has a temperate monsoon climate, hot and humid summer, dry and cold winter, average annual temperature of 12 °C and average annual precipitation of 800 mm. Rongcheng has convenient transportation, rich mineral resources and developed marine industries such as aquaculture and fishing. It has beautiful scenery and a national ocean park and a forest park, suitable for residential vacation. Seagrass houses are widely distributed in Rongcheng City, among which Dongchu Island Village on Ningjin Street is one of the most well-preserved areas in Jiaodong Peninsula. According to the planning document of Rongcheng City Rural Construction Plan (2017–2030), this paper selects 17 villages in Rongcheng City as sample, and the geographical location is shown in Figure 3.

Rongcheng has gone through the era of planting industry and fishery and is in the middle and late stages of industrialization and the period of great development of manufacturing industry. It is expected to enter the post-industrialization period around 2030. In recent years, Rongcheng has attached great importance to the protection, inheritance and development of traditional villages and actively explored new models for the protection of traditional villages. The protection and development of traditional villages have shown a good momentum. The list of “2022 Demonstration Counties (Cities, Districts) for the Centralized Protection and Utilization of Traditional Villages” was published on the websites of the Ministry of Finance and the Ministry of Housing and Urban–Rural Development of the People’s Republic of China. At present, there are 13 national traditional villages and 21 provincial traditional villages in Rongcheng. In the southern section of Ningjin Street, Dongchu Island Village, Dongdun Village, Liu Village, Malanjiang Village, Quge Village, Dayu Island Village and Renhe Town Courtyard Village, the common characteristics of traditional villages are fully excavated and integrated, and the “Shili Ancient Township” is formed.

Dazhuang Xujia, Dongyandun, Yandunjiao, Donggu Village of Northern Lidao Town, Xiaoxi Village and Weiwei Village of Gangxi Town have stacked seaweed houses, large swans in groups, forming a unique coastal fishermen scenery. In the concentrated protection of traditional villages, Rongcheng attaches great importance to the protection and restoration of traditional buildings and traditional features with historical and cultural values, highlighting cultural characteristics, and will also focus on meeting the needs of modern production and life, improving infrastructure and enhancing the vitality of traditional villages. According to a survey on the current situation of seagrass houses in Rongcheng City in 2018, there are 63 well-preserved traditional villages with seagrass houses. Among them, 36 are located in Xidao Town, 6 in Gangxi Town, 17 on Yongjin Street and 4 on Xunsan Street. The protection and utilization status of village seagrass houses: the seagrass houses are idle and abandoned, accounting for 30.2% of the total. In 52 villages, the seacoast is used as usual without special protection measures, accounting for 82.5 percent of the total. In addition, 14 villages including Xiangjiazhai have developed tourism and service industries, accounting for 22.2% of the total. Nakaido village exists in the way of museum protection. Finally, 21 villages have other conditions, accounting for 33.3% of the total.

### 3.2. Data Sources

The data used in this paper include high-resolution remote sensing image data, UAV flight data, 3D laser scanning data, auxiliary analysis data, etc. The data used in the article are from 2018. The remote sensing image data are GF-1 remote sensing satellite data of Rongcheng City, and the remote sensing image data are processed by projection transformation and frame cutting [39,40]. The UAV flight data are the POS point data and the original photos of the two characteristic villages of Dongchudao Village on Ningjin Street and Muyunan Village on Gangwan Street. After data processing such as distortion correction, aerial triangulation and image mosaic, 0.1 m resolution images are synthesized.

The three-dimensional laser data are the real data of the scanned seagrass houses. In the later stage, the data are processed and modeled to achieve the effect of restoring the real scene. Three-dimensional laser scanning is the first step to build the model, and the scale parameters of the buildings obtained need to be reconstructed using specialized modeling software, which is the second step of modeling. The modeling software used in this paper is SKETCHUP 2018; the Chinese name is Sketchup Master. It has a wide user base all over the world and can easily and quickly build house models. The last step in building the seagrass house model is the texture mapping of the seagrass house model surface.

Auxiliary analysis data include DEM data, population and socio-economic data, land-use data and various types of planning data and so on. The DEM data are downloaded from the website (http://www.resdc.cn (accessed on 15 February 2018)); population, GDP and other socio-economic data are mainly obtained from the statistical yearbook and field survey of Rongcheng City (Table 1). They are used to analyze the economic development status and housing-bearing capacity of characteristic villages in Rongcheng City, and they have an important reference for the evaluation of characteristic buildings. The planning data show the development direction of the region and relevant policies issued by the state, etc., which can help this paper evaluate and regionalize the characteristic style of the coastal region scientifically and reasonably. The planning data used in this paper mainly include general planning of land use in Rongcheng City, rural construction planning in Rongcheng City, etc. All kinds of measured data, planning data and population, social and economic data are statistically analyzed to provide basic data for quantitative evaluation of characteristic features.

### 3.3. Methods

#### 3.3.1. Construction of Evaluation Index System

Based on the analysis of the research on the evaluation index system of characteristic style, according to the selection principle of evaluation index, it is determined that the evaluation index system should reflect the value characteristics of rural house [11,12]. The characteristic styles of coastal rural house are evaluated from three aspects: the overall village environment, the value of coastal building and traditional folk culture. The overall village environment reflects the pattern and morphological structure of the village, including environmental style and overall pattern, and the spatial form of streets and courtyards. The former selects three indices: the beauty of the village natural environment, the preservation degree of the traditional village pattern and the longevity of the existing village. The latter selects two indices: the total length of traditional streets and the total area of traditional courtyards. The coastal building value includes two parts: historical value and characteristic value. The former is evaluated by the grade of cultural relic protection unit and the earliest building age of seagrass houses, while the latter is evaluated by three indices: the area percentage of seagrass houses to village buildings, the quantity percentage of seagrass houses and percentage of seagrass houses in good quality. Traditional folk culture includes three parts: the uniqueness of traditional folk culture, the maintenance degree of traditional folk skills and the persistence degree of traditional life mode. The quantity and permanence of customs, festivals, legends and poems, the quantity and permanence of handicrafts, traditional arts and performance forms, the proportion of the population living in seagrass houses to the permanent population and the proportion of the population engaged in traditional production methods such as agriculture and fishery are selected.

The evaluation index system is divided into target layer A, comprehensive evaluation layer Bn, factor evaluation layer Cn and subfactor evaluation layer Dn. After determining the target layer A, that is, the evaluation factor set and evaluation level of the protection of rural houses’ characteristic style, the comprehensive evaluation layer Bn is selected, and the factor evaluation layer Cn is determined on this basis. The subfactor evaluation layer Dn is determined under the control of factor evaluation layer Cn, and the evaluation index system of rural house characteristic style is established (Figure 2). On the basis of determining all of the evaluation indices, the sample data of 17 villages are prepared. Evaluation index system and data sources are shown in Table 1.

#### 3.3.2. Construction of Regionalization Index System

Through the differentiation and targeted evaluation and division of rural house features in Rongcheng City, the development conditions and directions of different villages are clarified so as to provide spatial guidance for the implementation of policies. In combination with the existing rural construction foundation of Rongcheng City and relying on landscape resources, the rural landscape pattern with clear zoning and distinctive features should be constructed. According to the goal of regionalization, the characteristic style regionalization of rural house should identify the characteristics and differences of villages so as to define different village types and carry out targeted protection, development and management. As mentioned above, the overall village environment is a comprehensive expression of rural site selection, village pattern, streets and courtyards, etc., reflecting the pattern and form of the countryside, while the characteristic style value of rural house is the sum of the tangible and intangible values of the whole village, mainly reflected through the building value and traditional cultural value, which is also the basis of regionalization. Village location and natural and socio-economic conditions are important factors for rural development, which not only affect the formation and current situation of the village but also affect the future development direction to a large extent. Regionalization provides spatial guidance for the implementation of differentiated management policies, so it is necessary to consider the protection, development and management of rural house characteristic style so as to make regionalization more targeted and practical.

Under the research of relevant scholars and the advice of experts [11,12], the regionalization of characteristic features of rural house should first consider the value of characteristic style of rural house, then combine the status quo of protection, development and management, further analyze the socio-economic conditions and location conditions of villages and finally consider the overall environment and natural conditions of villages so as to comprehensively identify different villages and type areas.

Taking 17 coastal villages in the coastal rural landscape area of Rongcheng City as the object, according to the regionalization objectives, following the corresponding principles and basis, referring to the existing research, on the basis of the evaluation of the characteristic style value of rural house, and comprehensively considering the rural location, social development, economic level, protection and development management status, the regionalization index system of rural house characteristic style is constructed (Figure 2), similarly including target layer A, comprehensive evaluation layer Bn, factor evaluation layer Cn and subfactor evaluation layer Dn, as shown in Table 2.

#### 3.3.3. Cluster Analysis

Cluster analysis is a method of establishing a classification that automatically classifies a batch of sample data (or variables) based on intimacy in the absence of prior knowledge. The basic idea of hierarchical clustering analysis is that each sample is independent at the beginning of clustering analysis. Then, according to the degree of intimacy between the samples, the most similar samples are clustered into a class of remaining samples and repeated until all the samples are clustered into a class. The calculation results of the regionalization indices of each village are obtained by weighting and summing the corresponding indices of each village with the determined weight of each index. With the help of SPSS 26.0 software, based on the calculation results of the regionalization indices of the characteristic style of rural houses in Rongcheng City, the Q-type clustering of 17 villages is carried out by using Euclidean square distance and system clustering method.

## 4. Results and Analysis

### 4.1. Evaluation of Coastal Rural Houses’ Characteristic Style

#### 4.1.1. Analysis of Comprehensive Evaluation Results

The evaluation value of each sample is obtained by the weighted summation of sample data, and the three types of indices and total scores of the comprehensive evaluation layer B of each village are calculated; that is, the evaluation results of the characteristic style of coastal rural houses of each village are shown in Table 3 and Figure 4.

It can be seen that among the three categories of indices in the comprehensive evaluation layer B of the characteristic style evaluation, the score of the coastal building value part accounts for a relatively high proportion of the total score of most villages and has a greater impact on the total score. There are two villages with a score of more than 60 points, which are Dongchu Island Village and Dazhuang Xujia Community. Among them, Dongchu Island Village ranks first, and the characteristic style of rural houses is the highest. A total of eight villages scored 50–60 points, namely Muyunan Village, Quge Village, Weiwei Village, Sohou Wangjia Village, Zhimatan Village, Dongyandun Community, Xiangjiazhai Village and Liu Village. There are seven villages that scored below 50 points: Malangou Village, Xiaoxi Village, Yandunjiao Community, Suhou Lujia Village, Donggu Village, Jiayuwang Village and Yuankuang Village.

#### 4.1.2. Analysis of Single-Factor Evaluation Results

According to the three indices of the overall village environment of the comprehensive evaluation layer B, the coastal building value and the traditional folk culture, the scores of each village are ranked, and the highest one is selected as the main characteristic of the village’s house style. The evaluation classification results are obtained according to the main characteristics of the different villages (Table 4).

It can be seen that Muyunan Village has the highest total score in the characteristic villages dominated by the overall village environment, and it is the village with the highest degree of rural house characteristics of this type. In the characteristic villages dominated by the coastal building value, the total score of Dazhuang Xujia Community is the highest, which is the village with the highest degree of a rural house characteristic of this type. It can be seen that there is no obvious regularity in the spatial distribution of villages with different characteristic styles. Natural conditions, locational conditions and socio-economic data should also be considered in subsequent regionalization studies.

### 4.2. Regionalization of Coastal Rural Houses’ Characteristic Style

According to the clustering results (Figure 5), the first category is Dongchudao Village, Malangou Village and Jiayuwang Village. The overall environment, the value of houses’ characteristic style and the status of protection, development and management of Dongchu Island Village are in good condition. The seagrass houses in this village are large in scale and concentrated in distribution. The oldest seagrass house has a history of more than 300 years, which essentially continues the historical features. It can be called a living specimen of domestic coastal ecological dwellings. Malangou Village and Jiayuwang Village are both located in coastal areas, facing the sea, and are typical coastal villages. The overall pattern of the village, building and road accessibility is good, and the village and the surrounding environment are natural. The leading industries in the villages are mariculture and planting.

Class II is Muyunan Village and Yuankuang Village. The value of the houses’ characteristic style and the status of protection, development and management are relatively good. In Muyunan Village, there are more than one hundred sets of traditional residential courtyards of the Qing Dynasty. Most of the buildings have the typical architectural features of the Rongcheng mountainous area. It is one of the traditional mountainous villages with well-preserved historical buildings in Shandong Province and is also a famous fisherman’s painting village. The social and economic conditions and the status of protection and development management of the Yuankuang village are higher. The external space environment is beautiful, and the landscape level is rich; the number of stone-tile houses is large, all over the village. A large part of them were built in the 1960s and 1970s. The stone houses with a history of more than 100 years are mainly concentrated in the middle of the village. The ceremony of fishermen’s sacrifice to the sea in the Grain Rain Festival is a provincial intangible cultural heritage. The leading industries of the village are marine fishing, mariculture and international and domestic transportation. The collective income of the whole village is nearly CNY 500 million, and the per capita income of farmers is about CNY 12,000. The village collective invests a large amount of funds and adopts various measures to protect, develop and manage the traditional culture of the village.

Class III: Dazhuang Xujia Community, Yandunjiao Community, Weiwei Village, Dongyandun Community, Donggu Village, Xiaoxi Village and Xiangjiazhai Village. The results of the regionalization indices are below the middle. Different villages have differentiated development advantages. Among them, the value of houses’ characteristic style of Dazhuang Xujia Community and Dongyandun Community are high. There are hundreds of traditional residential courtyards of seaweed houses from the Ming and Qing Dynasties in the village. Most of the buildings appeared in the Qin and Han Dynasties and flourished in the Ming and Qing Dynasties. The seaweed house in Yandunjiao Village is a typical one of Rongcheng and is distributed intensively. The oldest seaweed house has a history of several hundred years. The collective economy is booming, with the major industries including fishmeal processing, shipbuilding and mariculture. The remaining villages are provincial-level traditional villages. Except for Xiaoxi Village, there are special protection plans. The status quo of protection, development and management is good. The traditional pattern in the traditional village gathering area in the village is relatively complete, and the traditional texture space is formed along the main streets.

Class IV: Liucun, Suohou Wangjiacun, Suohou Lujiacun, Zhimatancun and Qugecun. The comprehensive score of the village is at the lowest level, and the results of each index layer are not dominant. The village has a long history, and the overall style has certain characteristics. There are some seagrass houses and ancient dwellings in the traditional village area. However, the village does not have a corresponding protection plan, farming and aquaculture are the main sources of livelihood, industrial development is relatively backward, and the village collective income and per capita income are low. These kinds of villages face problems such as the insufficient protection of traditional buildings in ancient villages, challenges to the living inheritance of folk culture and imperfect service facilities.

Based on the calculation results of the regionalization indices, according to the administrative ownership of 17 villages and the relevant development plans of Rongcheng City, especially the Rural Construction Plan of Rongcheng City (2017–2030) (in terms of the tourism radiation area, 17 villages can be attributed to the tourism space layout of “one center, two wings and two corridors” in Rongcheng City (Tourism Planning of Rural Construction in Rongcheng City (2017–2030))), it is ensured that villages within the radiation range of adjacent tourist areas or administrative areas are divided together to facilitate later protection and development management. Then, according to the characteristics of the village, with reference to the characteristic village style guidance type and beautiful rural area planning, the characteristic style regionalization of rural houses is determined (Figure 6, Table 5).

## 5. Discussion

### 5.1. The Relationship between Characteristic Style Development Direction of Rural Houses and the Rural Revitalization

The protection of traditional villages allows us to retain the villagers, protect the countryside and remember homesickness. Through traditional villages, an evolving framework of rural revitalization and cultural routes can be established, in terms of the inherent properties of both (Figure 7). First, among the four major types of rural revitalization promotion, the feature conservation type essentially refers to traditional villages with historical value and cultural heritage. Second, traditional villages constitute the elements of cultural routes from points to lines. These villages as elements must share the associated characteristics of cultural routes in three ways. Finally, cultural routes have striking potential to promote rural revitalization from line to region, through an evolving strategy [41]. As an important part of the rural regional system, traditional villages are the main positions of rural revitalization. The transformation and development of their living environment is a stress response under the national strategic blueprint such as beautiful villages and new urbanization and the only way to comply with the requirements of the times and the objective laws of villages [42].

It should also promote protection through utilization, so as to realize the active inheritance of material and intangible traditional culture and enhance the endogenous impetus of the protection and development of traditional villages. The natural landscape, historical culture, idyllic scenery and other resources of traditional villages should be fully explored, and rural tourism, cultural creativity and other industries should be developed according to local conditions. It has greatly increased the income of local farmers, promoted rural revitalization, enhanced cultural confidence and brought new vigor and vitality to traditional villages. Traditional villages are living history. To protect traditional villages, it is necessary to preserve their original features and meet the needs of the villagers for a better living environment. Therefore, in the process of protecting traditional villages, it is necessary to improve the lives of the villagers and make the protection of traditional villages more reliable, so as to realize the living inheritance of the material and intangible traditional culture.

The demonstration of the centralized protection and utilization of traditional villages will form a number of protection and utilization paths and methods for traditional villages with different types and characteristics and produce replicable experiences and modes of protection and utilization of traditional villages in the process of protection and utilization, further promoting the in-depth development of the protection and utilization of traditional villages. It would also offer the following benefits: an upgrade of the protection and utilization of cultural resources in traditional villages; a fundamental change in the overall environment of the village; promotion of the integrated development of “culture +” in traditional villages; an increase in the income of village residents; and promotion of common prosperity. In addition, it should integrate policy resources and social funds, make use of characteristic rural resources and form effective ways to promote comprehensive rural revitalization through the protection and utilization of traditional villages (Figure 7).

With the characteristics of mountain and sea culture as the core, Rongcheng will be built into a leading area of rural construction in Shandong Province and a demonstration area for the overall construction and development of towns and villages by realizing the reasonable allocation of the diversified rural population, social economy and resource elements, perfecting the construction of rural public services and municipal infrastructure and improving the rural production and living environment.

### 5.2. Policy Recommendations

Each region should be combined with the existing rural construction foundation, and the style shape should highlight the “characteristic”, that is, highlight traditional characteristics, create regional characteristics, highlight heritage national characteristics and the cultural characteristics of organic unity and adjust measures to local conditions [39]. Relying on the rich landscape resources of mountains and rivers, the following rural characteristics with obvious landscape division and remarkable landscape characteristics are constructed:(1)Historical culture + industrial development characteristic area. This region includes Dongchu Island Village, Malangou Village and Jiayuwang Village, located in the eastern coastal areas of Rongcheng City. The traditional houses, ancient trees, antiquities and coastline scenery, modern fishery, ecological agriculture and industrial tourism combine leisure tourism and cultural heritage. Dongchu Island Village builds a marine historical and cultural display base through a rural memory hall, fisherman life experience, seagrass house living experience, etc. (Figure 8). Malangou Village uses the fisherman station and swan lake park construction to create a waterfront customs history display village. Jiayuwang Village is a key cultural tourism brand in Rongcheng City, integrating red tourism, ecological sightseeing, folk experience and cultural industry.(2)Folk customs + industrial development characteristic area. This area includes Yuankuang Village and Muyunan Village, located in the south of Rongcheng City. The beach shoreline resources of the coastal villages in the village are rich, the area of seawater aquaculture is large, and the number of fishing ports and docks is large. It is combined with good leisure projects, industrial tourism, the Shawo fishing port and the Chashan scenic spot, tasting delicious seafood, purchasing high-quality local products, viewing mountain and sea scenery and experiencing farming, sea and animal husbandry fish scenery. Muyunan Village is located in the southeast coastal area of the Chishan Scenic Area. With the excellent natural landscape resources and profound historical and cultural heritage of the Chishan Scenic Area as the core and the village’s own unique painting and calligraphy culture as the divergence point, it complements the Chishan Scenic Area in function and landscape and has become an important leisure and vacation comprehensive service area in the south of Rongcheng.(3)Natural scenery characteristic area. The region is located in the Good Luck Corner Resort on the northeast coast of Rongcheng City, which is covered by seven sample villages: Yandunjiao Community, Donggu Village, Xiaoxi Village, Weiwei Village, Dongyandun Community, Dazhuangxujia Community and Xiangjiazhai Village, mainly including the three towns of Gangxi, Chengshan and Lidao. The villages in the area have a certain natural foundation and form an area with Beihai scenery, Wanmu Forest Park, Xixiakou Village and scenic spots. Relying on the advantageous resources such as the Naxianghai Scenic Area, Chengshan Forest Farm Leisure Resort, Jiming Island, Hekou Fishing Village, Datian E National Nature Reserve, Chaoyang Port and Mashan to make use of the sea and increase interest and extend the tourist routes of Xixiakou Scenic Area, the combination of ecological protection, sightseeing photography tourism and leisure fishery is adopted. Thus, the characteristics of the village and the planning area are taken together, from all aspects of the pastoral dream, rural dream and the future dream as a planning foothold.(4)Folk customs characteristic area. This area includes Quge Village, Zhimatan Village, Liu Village, Suohou Wangjia Village, Suohou Lujia Village, etc., and is located in the Shidao Bay Tourist Resort in the southeast of Rongcheng. Through the cultivation of characteristic projects, the greening in and around the village and in courtyards is strengthened, enhancing the village environment and vigorously creating pastoral scenery. Using agricultural planting and breeding, tourist experience activities should be carried out, and through the experience of agricultural production activities, the purpose of farming experience should be achieved.

**Figure 8 ijerph-20-03010-f008:**
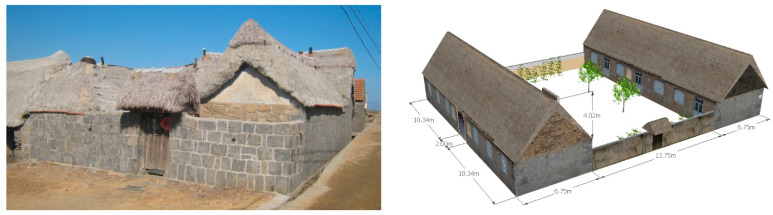
Seagrass room.

Through the regionalization of the characteristic style of rural houses, the development conditions and construction directions of different villages are clarified. Combined with the regional development orientation, the guarantee support highlights “vitality”, innovates the mechanism and stimulates vitality. The protection and improvement of the characteristic style of rural houses is an important part of the construction of beautiful countryside, and it is also an essential requirement of rural ecological livability. Government departments should formulate relevant laws and regulations from the aspects of protection, inheritance and development and form mandatory social norms. The competent government departments should compile the technical guidelines for the guidance of the characteristics of rural residential buildings, guide the residential buildings to be ecologically livable and distinctive in the characteristics shaping and protection and effectively inherit and protect the historical art and scientific value of residential buildings. Overall planning, construction and management should accelerate the scientific, orderly and sustainable development of the protection and promotion of rural residential features. The protection and promotion of residential features with the development of rural industries should be combined to construct a development and protection system. Local governments should strictly enforce the relevant laws and regulations, on the basis of town and village planning, to improve the characteristics of rural residential planning and incorporate it into the town and village planning implementation. The “one village, one product” and “one village, one rhyme” characteristics of rural houses should be gradually formed, and the authenticity of rural houses, integrity and continuity should be effectively retained (Figure 7).

The development of rural areas is an important foundation for rural revitalization and even China’s economic development. The protection and development management of residential features in traditional villages is an important part of rural development. The characteristic style of rural houses embodies the historical and cultural value of the village. Through a quantitative evaluation and regionalization study, the development conditions and directions are defined, and corresponding measures are set up to protect and enhance the value of the style. On the one hand, this can be connected to the rural revitalization strategy, and it is more complementary to the construction of beautiful China; on the other hand, this is also the specific path to implement the “Weihai Beautiful Village Planning and Design Technical Guidelines” and the “Rongcheng Urban Rural Construction Plan (2017–2030)”. This will not only promote the protection and construction of the residential features of coastal traditional villages but also provide reference directions and measures for the employment of village industries and farmers, the long-term stable increase in local people’s income and living and working in peace and contentment.

## 6. Conclusions and Research Prospects

### 6.1. Conclusions

This study evaluates the characteristics of coastal rural houses in 17 villages in the coastal area of Rongcheng and constructs an evaluation and regionalization index system suitable for the region. Taking the overall village environment, the coastal building value and the traditional folk culture as the comprehensive evaluation layer factor B, combined with the previous research and the collected data, the evaluation results of the characteristic style of coastal rural houses are calculated.

According to the results obtained, the characteristic style of the current residential features of each village is obtained. Among all villages, Dongchudao Village ranks first, and the characteristic style of rural houses is the highest. At the same time, the dominant types of the characteristic style of each village are classified into three types: the characteristic style dominated by the overall village environment, the characteristic style dominated by the coastal building value and the characteristic style dominated by traditional folk culture.

Coastal rural houses’ characteristic style of Rongcheng can be divided into four consecutive areas: Dongchudao Village as the representative of the historical and cultural + industrial development characteristics of the landscape area, Muyunan Village and Yuankuang Village as the representatives of the folk customs + industrial development characteristics of the landscape area, Xiaoxi Village and Yandunjiao Community as the representatives of the natural scenery of the landscape area and Quge Village and Zhimatan Village as the representatives of the folk customs of the landscape area. In accordance with the requirements of the development plan, according to different rural conditions, the construction direction of different regional rural residential features put forward the protection and promotion measures.

### 6.2. Research Limitations and Prospects

In terms of data processing, this paper uses a combination of remote sensing image data, UAV flight data and seagrass house 3D laser scanning data to fully display the spatial information of seagrass house buildings. However, due to the low efficiency of 3D laser scanning data acquisition, the 3D model of seagrass houses cannot be obtained in a large area, so the data acquisition method and efficiency need to be improved. At the same time, future research should address the lack of socio-economic data and historical data collection.

As the coastal rural residential features are an organic complex, the evaluation index also has considerable complexity. Restricted by data acquisition, there are some choices in selecting indices when constructing an evaluation index system. In the later stage, we can continue to strengthen the collection of data on the characteristics of rural residential features and further optimize the evaluation index system of coastal rural residential features.

## Figures and Tables

**Figure 1 ijerph-20-03010-f001:**
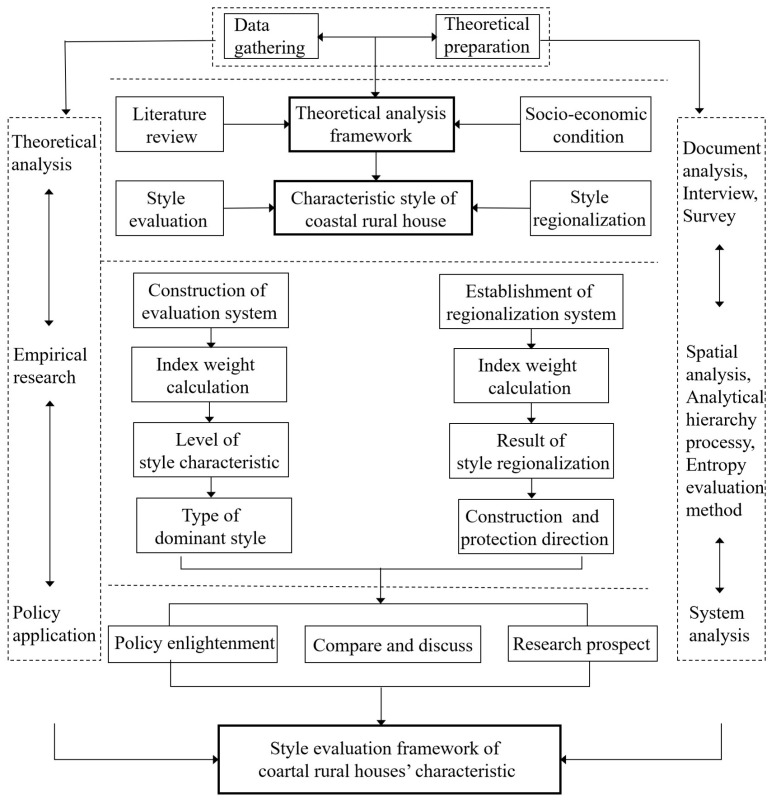
Overall flow chart of the method.

**Figure 2 ijerph-20-03010-f002:**
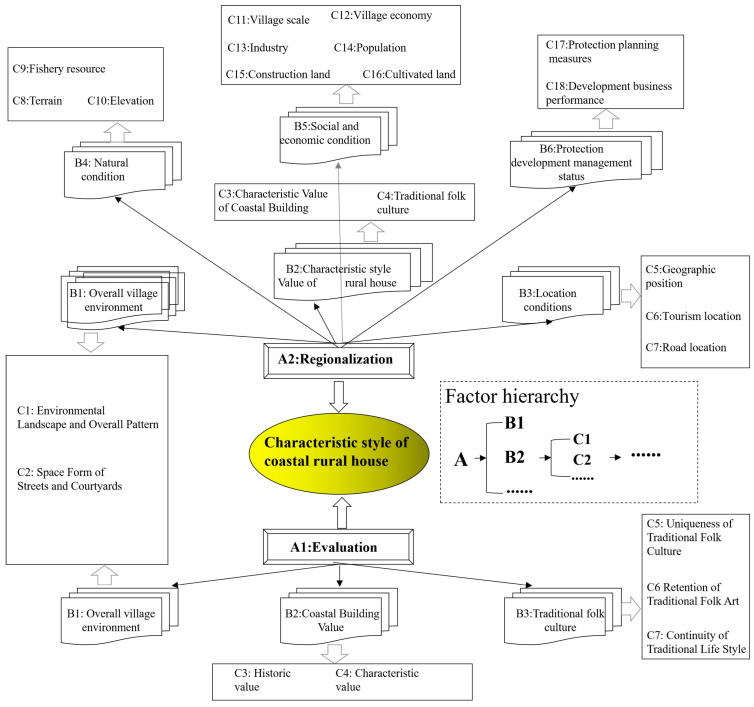
Frame construction diagram of indicator system of rural houses’ characteristic style.

**Figure 3 ijerph-20-03010-f003:**
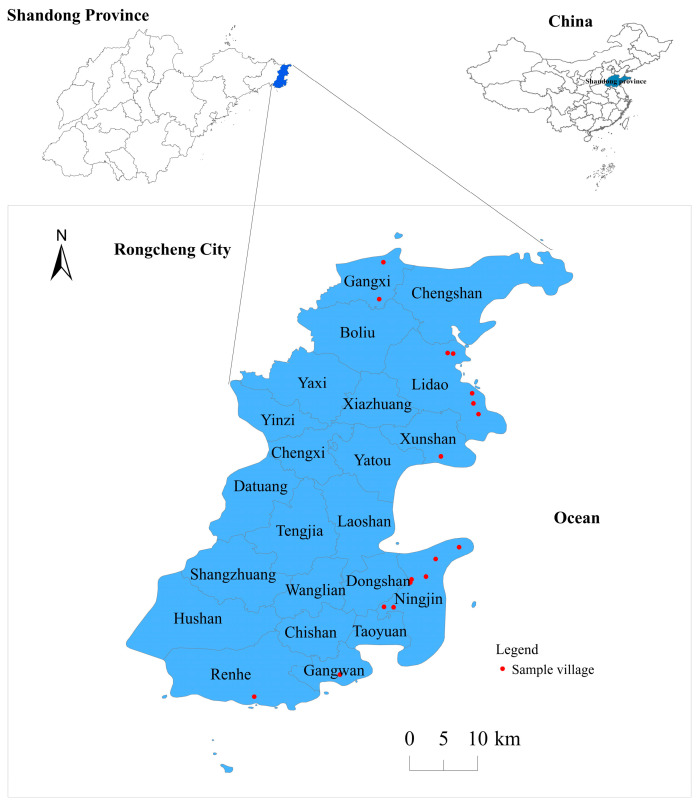
Study area.

**Figure 4 ijerph-20-03010-f004:**
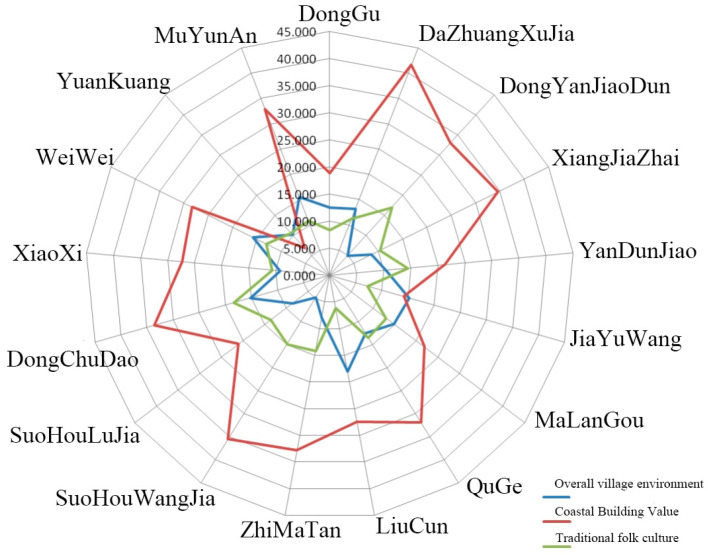
Evaluation results of rural houses’ characteristic style.

**Figure 5 ijerph-20-03010-f005:**
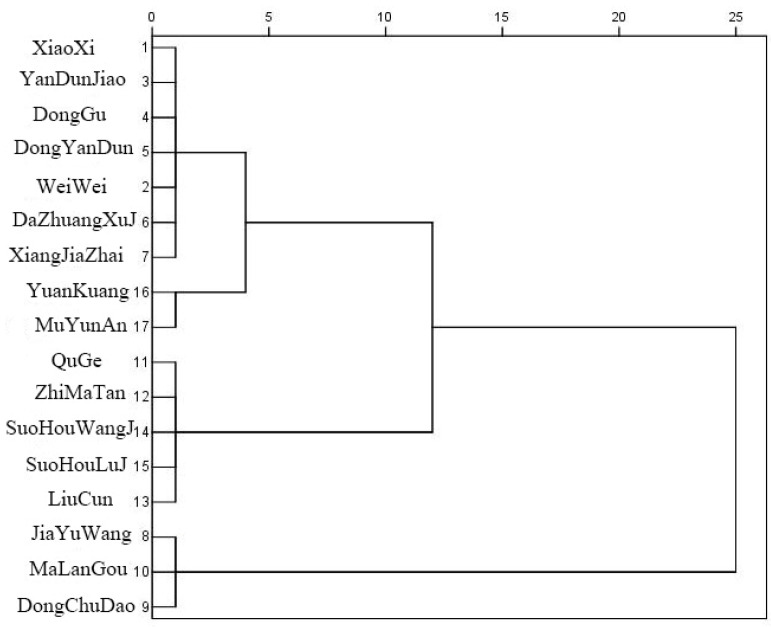
Clustering pedigree diagram of rural houses’ characteristic style in Rongcheng City.

**Figure 6 ijerph-20-03010-f006:**
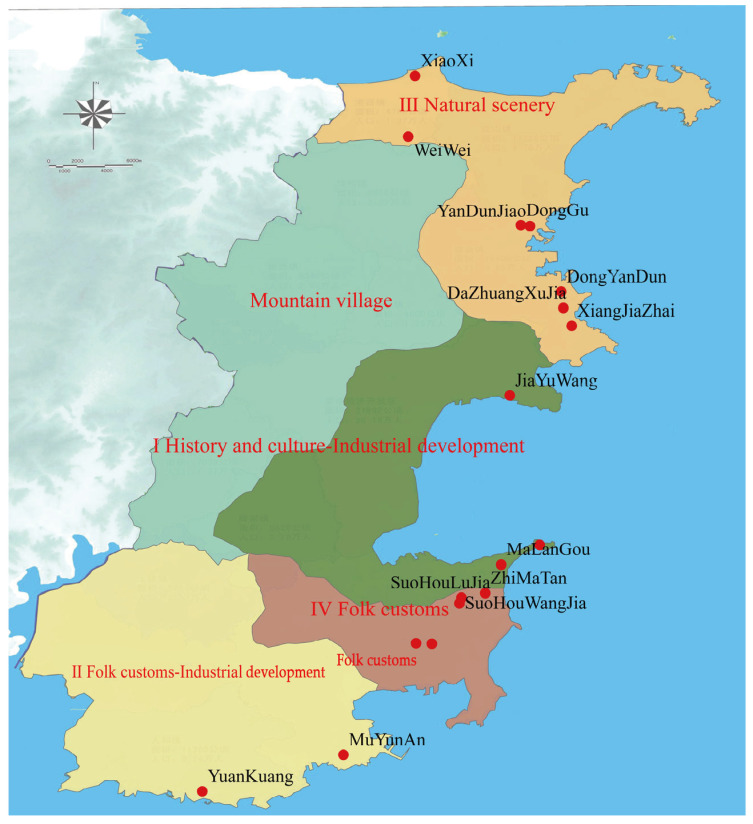
Rural characteristic style regionalization of Rongcheng City.

**Figure 7 ijerph-20-03010-f007:**
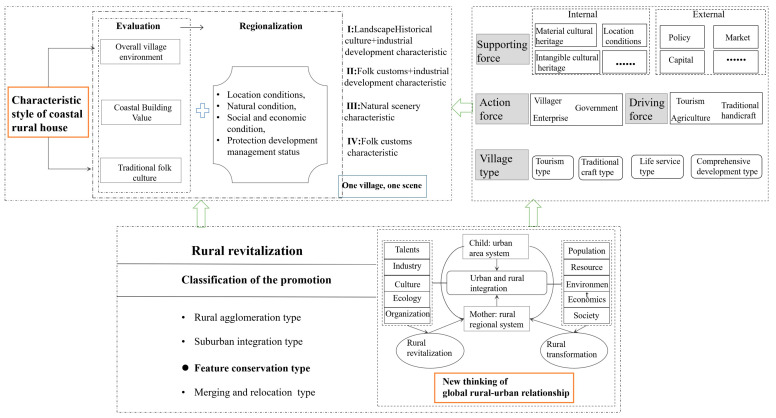
Interaction mechanism between characteristic style development direction of rural houses and the rural revitalization.

**Table 1 ijerph-20-03010-t001:** Evaluation index framework system.

Target Layer (A)	Comprehensive Evaluation Layer (B, Weight)	Factor Evaluation Layer (C, Weight)	Subfactor Evaluation Layer (D, Weight)	Data Sources	Weight
Evaluation Factors of Rural Residential Features A	B1 Overall village environment (0.25)	C1 Environmental Style and Overall Pattern (0.50)	D1 The beauty of village natural environment (0.52)	Field research	0.065
			D2 Preservation degree of village traditional pattern (0.24)	Field research	0.030
			D3 Longevity of the existing village (0.24)	Village archives	0.030
		C2 Space Form of Streets and Courtyards (0.50)	D4 Total length of traditional streets (0.543)	UAV data + 3D laser scanning data	0.068
			D5 Total area of traditional courtyard (0.457)	UAV data + 3D laser scanning data	0.057
	B2 Coastal building value (0.5)	C3 Historic Value (0.25)	D6 Grade of cultural relics protection unit (0.82)	Village archives	0.102
			D7 Earliest building age of seagrass houses (0.18)	Village archives	0.023
		C4 Characteristic Value (0.75)	D8 The area percentage of seagrass houses to village buildings (0.43)	Unmanned aerial vehicle data	0.161
			D9 The quantity percentage of seagrass houses (0.224)	Village archives	0.084
			D10 Percentage of seagrass house in good quality (0.346)	Village archives + field research	0.130
	B3 Traditional folk culture (0.25)	C5 Uniqueness of Traditional Folk Culture (0.25)	D11 The quantity of customs, festivals, legends and poems (0.531)	Village archives + field research	0.033
			D12 The permanence of customs, festivals, legends and poems (0.469)	Village archives + field research	0.029
		C6 The Maintenance Degree of Traditional Folk Skills (0.25)	D13 The quantity of handicrafts, traditional arts and performance forms (0.551)	Village archives + field research	0.034
			D14 The permanence of handicrafts, traditional arts and performance forms (0.449)	Village archives + field research	0.028
		C7 The Persistence Degree of Traditional Life Mode (0.50)	D15 The proportion of the population living in seagrass houses to the permanent population (0.55)	Village archives + field research	0.069
			D16 The proportion of the population engaged in traditional production methods such as agriculture and fishery (0.45)	Village archives + field research	0.057

**Table 2 ijerph-20-03010-t002:** Index framework system for regionalization.

Target Layer (A)	Comprehensive Evaluation Layer (B, Weight)	Factor Evaluation Layer (C, Weight)	Subfactor Evaluation Layer (D, Weight)	Weight
Characteristic style regionalization of rural house	Overall village environment (0.15)	Environmental style and overall pattern (0.5)	The beauty of village natural environment (0.52)	0.039
			Preservation degree of village traditional pattern (0.24)	0.018
			Longevity of the existing village (0.24)	0.018
		Space form of streets and courtyards (0.5)	Total length of traditional streets (0.54)	0.040
			Total area of traditional courtyard (0.46)	0.035
	Value of characteristic style of rural house (0.31)	Characteristic value of coastal building (0.5)	Grade of cultural relics protection unit (0.34)	0.053
			Earliest building age of seagrass houses (0.08)	0.012
			The area percentage of seagrass houses to village buildings (0.25)	0.039
			The quantity percentage of seagrass houses (0.13)	0.020
			Percentage of seagrass house in good quality (0.20)	0.031
		Traditional folk culture (0.5)	The quantity of customs, festivals, legends and poems (0.36)	0.056
			The permanence of customs, festivals, legends and poems (0.28)	0.043
			The quantity of handicrafts, traditional arts and performance forms (0.13)	0.020
			The permanence of handicrafts, traditional arts and performance forms (0.11)	0.017
			The proportion of the population living in seagrass houses to the permanent population (0.06)	0.009
			The proportion of the population engaged in traditional production methods such as agriculture and fishery (0.05)	0.008
	Location conditions (0.13)	Geographic position (0.57)	Latitude (1.00)	0.074
		Tourism location (0.29)	Distance to tourist attractions (1.00)	0.038
		Road location (0.14)	Distance to main road (1.00)	0.018
	Natural condition (0.06)	Terrain (0.25)	Plain better than hill better than mountain (1.00)	0.015
		Fishery resources (0.50)	Aquaculture area (1.00)	0.03
		Elevation (0.25)	Elevation above sea level (1.00)	0.015
	Social and economic condition (0.1)	Village scale (0.1)	Village area (1.00)	0.010
		Village economy (0.3)	Gross income (1.00)	0.030
		Industry (0.3)	Proportion of primary industry (1.00)	0.030
		Population (0.1)	Population size (1.00)	0.010
		Construction land (0.1)	Construction land area (1.00)	0.010
		Cultivated land (0.1)	agricultural acreage (1.00)	0.010
	Protection development management status (0.25)	Protection planning measures (0.5)	Protection planning integrity (0.34)	0.043
			Implementation degree of protection planning (0.66)	0.083
		Development business performance (0.5)	Infrastructure integrity (0.27)	0.034
			Income per capita (0.73)	0.092

**Table 3 ijerph-20-03010-t003:** Comprehensive evaluation results of rural houses’ characteristic style.

Number	Name	Overall Village Environment	Coastal Building Value	Traditional Folk Culture	Aggregate Score
1	DongChuDao	15.262	33.573	18.376	67.211
2	DaZhuangXuJia	13.101	41.648	11.265	66.015
3	MuYunAn	15.468	32.925	10.776	59.169
4	QuGe	12.454	31.946	13.442	57.842
5	WeiWei	15.829	28.246	13.094	57.168
6	SuoHouWangJia	4.805	35.552	14.852	55.209
7	ZhiMaTan	8.003	32.834	14.116	54.953
8	DongYanDun	4.858	33.001	16.962	54.822
9	XiangJiaZhai	8.539	34.618	10.385	53.542
10	LiuCun	17.996	27.502	6.115	51.613
11	MaLanGou	14.839	21.873	13.036	49.749
12	XiaoXi	9.188	27.319	10.679	47.186
13	YanDunJiao	10.421	21.298	14.440	46.159
14	SuoHouLuJia	8.620	21.014	13.542	43.176
15	DongGu	12.532	18.772	8.342	39.646
16	JiaYuWang	15.223	14.164	7.160	36.547
17	YuanKuang	10.184	6.847	10.593	27.624

**Table 4 ijerph-20-03010-t004:** Single-factor evaluation results of rural houses’ characteristic style.

Name	Total Score Ranking	Village Overall Environment Score Ranking	Coastal Building Value Score Ranking	Traditional Folk Culture Score Ranking	Characteristic Style Type
DongChuDao	1	4	4	1	Traditional Folk Culture Oriented
DaZhuangXuJia	2	7	1	10	Coastal Building Value Oriented
MuYunAn	3	3	6	11	Whole Village Environment Leading Type
QuGe	4	9	8	7	Traditional Folk Culture Oriented
WeiWei	5	2	9	8	Whole Village Environment Leading Type
SuoHouangJia	6	17	2	3	Coastal Building Value Oriented
ZhiMaTan	7	15	7	5	Traditional Folk Culture Oriented
DongYanDun	8	16	5	2	Traditional Folk Culture Oriented
XiangJiaZhai	9	14	3	14	Coastal Building Value Oriented
LiuCun	10	1	10	17	Whole Village Environment Leading Type
MaLanGou	11	6	12	9	Whole Village Environment Leading Type
XiaoXi	12	12	11	12	Coastal Building Value Oriented
YnaDunJiao	13	10	13	4	Traditional Folk Culture Oriented
SuoHouLuJia	14	13	14	6	Traditional Folk Culture Oriented
DongGu	15	8	15	15	Whole Village Environment Leading Type
JiaYuWang	16	5	16	16	Whole Village Environment Leading Type
YuanKuang	17	11	17	13	Whole Village Environment Leading Type

**Table 5 ijerph-20-03010-t005:** Characteristic style regionalization of rural house in Rongcheng City.

Type	Villages	Township (Street)	Tourism Spatial Pattern	Division Name
Ⅰ	DongChuDao	NingJin	Chu Dao Resort	Historical and cultural characteristics type
	MaLanGou	NingJin	Chu Dao Resort	Historical and cultural characteristics type
	JiaYuWang	XunShan	Jiayuwang Tourist Resort	Historical and cultural characteristics type
Ⅱ	MuYunAn	GangWan	Chishan Scenic Area	Folk customs + industrial development characteristic type
	YuanKuang	RenHe	Chashan Scenic Area	Folk customs + industrial development characteristic type
Ⅲ	XiaoXi	GangXi	Good Luck Corner Tourist Resort	Natural scenery characteristic type
	WeiWei	GangXi	Good Luck Corner Tourist Resort	Natural scenery characteristic type
	YanDunJiao	LiDao	Good Luck Corner Tourist Resort	Natural scenery characteristic type
	DongGu	LiDao	Good Luck Corner Tourist Resort	Natural scenery characteristic type
	DongYanDunJiao	LiDao	Good Luck Corner Tourist Resort	Natural scenery characteristic type
	DaZhuangXuJia	LiDao	Good Luck Corner Tourist Resort	Natural scenery characteristic type
	XiangJiaZhai	LiDao	Good Luck Corner Tourist Resort	Natural scenery characteristic type
Ⅳ	QuGe	NingJin	Shidao Bay Tourism Resort	Folk customs characteristic type
	ZhiMaTan	NingJin	Shidao Bay Tourism Resort	Folk customs characteristic type
	LiuCun	NingJin	Shidao Bay Tourism Resort	Folk customs characteristic type
	SuoHouWangJia	NingJin	Shidao Bay Tourism Resort	Folk customs characteristic type
	SuoHouLuJia	NingJin	Shidao Bay Tourism Resort	Folk customs characteristic type

## Data Availability

Not applicable.

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
