# Peer review of "Developing an Evaluation System Suitable for Coastal Rural Houses’ Characteristic Style and Its Inspiration for Rural Revitalization: Case Study of Rongcheng in Shandong Province"

_ijerph, 2023, doi:10.3390/ijerph20043010_

Round 1

Reviewer 1 Report

Rural residential landscape is one of the important cultural landscapes,and is also an important embodiment of traditional rural culture. Protecting and inheriting the characteristic traditional residence culture is of great significance to the construction of beautiful countryside. The coastal village is an important part of characteristic countryside. However, it has not received sufficient attention from the geographical community, so far. The study evaluated the rural houses' characteristics in coastal villages, and  classifies the dominant types of villages' distinctive features. It 's very imteresting, and It is of great value to understand coastal rural culture. But there are some improvements before publish. 

      (1) There are many studies on ethnic minority villages, which are not considered in the literature review. 

    (2) in section 1 , the authors claim to adopt subjective and objective weighting method in line 106. However, we cannot find the introduction of this method in the second part or even the full text.

(3) in section 2.1 , the  description of the basis and reasons for selecting these indicators are too simple.

(4) in  Table2-1, we cannot know the data source of D3. In addition, the author should clarify the year of the study, the temporal attributes of the data.

(5) in Line 135, "referring to the existing research", what's the existing research, which, when, and who?

(6) in line 151, the calculation results of the zoning indexes of each village are obtained by weighting and summing the corresponding indexes of each village with the determined weight of each index.  we cannot know how is the weight calculated.

Author Response

List of Changes

First of all, we appreciate your careful review and positive comments! All of your comments are very important and really helpful to revise and improve our manuscript (MS). According to your comments, we have made relevant changes (words in red) to the whole MS.The detailed revisions are listed below, responding to your comments point by point.

  1. 1. Replies to reviewer #1

Rural residential landscape is one of the important cultural landscapes,and is also an important embodiment of traditional rural culture. Protecting and inheriting the characteristic traditional residence culture is of great significance to the construction of beautiful countryside. The coastal village is an important part of characteristic countryside. However, it has not received sufficient attention from the geographical community, so far. The study evaluated the rural houses' characteristics in coastal villages, and  classifies the dominant types of villages' distinctive features. It 's very interesting, and It is of great value to understand coastal rural culture. But there are some improvements before publish. 

      Point 1: There are many studies on ethnic minority villages, which are not considered in the literature review. 

Response 1: Thank you very much for valuable comments. 

We've already supplemented references to international and theoretical literature on ethnic minority villages issues listed by you. For example:

“12.Zhang, Z.;Yang, Q.; Wang, L.; Su, K.; Kuang, C. Traffic accessibility analysis of traditional villages in minority areas:a case study of Tongren City, Guizhou Province. Resources Science, 2018,40(11),2296-2306. 

13.Li, B.; Xu,C.; Zheng,S.; Wang,S.;Dou, Y. Spatial layout characteristics of Ethnic Minority villages based on pattern language: A case study of Southern Dong Area in Xiangxi as an example. Scientia Geographica Sinica,2020,40(11), 1784-1794.

14.Qin, X.; Li, X.; Chen, W.; Zhang, X. Spatial pattern and influenceing factors of ethnic minority villages in the yangtze river economic belt. Huanman geography, 2022,37(3),118-130.”

    Point 2: in section 1 , the authors claim to adopt subjective and objective weighting method in line 106. However, we cannot find the introduction of this method in the second part or even the full text.

Response 2:We explained it further. There are mainly subjective weighting method and objective weighting method to determine the weights of evaluation indicators. Subjective and objective weighting methods have their own advantages, but they also have certain defects and deficiencies. The weights determined by subjective weighting method can reflect the intention of researchers, but human factors limit the application of actual data. However, the weights determined by objective weighting method are closely related to actual data, but are susceptible to extreme values.

Therefore, this study uses a comprehensive weight determination method combining subjective and objective weighting methods, and chooses the classical subjective weighting method -- analytic hierarchy process (AHP) and the objective weighting method reflecting actual data -- entropy value method. Combining the two methods can complement each other and reduce the limitations of a single method. The evaluation index system constructed in this study consists of four layers. For the comprehensive evaluation layer and the factor evaluation layer, the analytic hierarchy process is used to calculate the weight of A, and for the factor evaluation layer, the entropy method is used to calculate the relative weight. The weight of each factor calculated by the two methods is multiplied to obtain the weight of the comprehensive subjective and objective weighting method.

Point 3: in section 2.1 , the  description of the basis and reasons for selecting these indicators are too simple.

Response 3: We added Theoretical underpinning as part 2. Among them, we carried out a theoretical analysis on the construction of the characteristic style and index system of coastal rural house, and referred to the research results of relevant scholars.

“The characteristic style of rural residential houses is formed based on certain geographical space, combined with characteristic historical and cultural factors. The overall village environment is a comprehensive expression of rural location, village pattern, streets and courtyards, and reflects the pattern and form of the village. The characteristic style value of rural dwellings is the sum of tangible and intangible values of the whole village, which is mainly reflected by architectural value and traditional cultural value. The main style of coastal residential in Rongcheng is that there are a large number of sea grass houses. The distribution and preservation status of sea grass houses in different villages are different, so each has its own characteristics. Therefore, this study needs to take into account the characteristics of coastal buildings in the quantitative style evaluation of coastal rural residential, while, combine with the characteristics of folk culture to build a suitable evaluation index system. Evaluation and zoning is to provide spatial guidance for the implementation of differentiated management policies, so it is necessary to consider the protection, development and management of rural residential features, so as to make it more targeted and practical........”.

Point 4: in  Table2-1, we cannot know the data source of D3. In addition, the author should clarify the year of the study, the temporal attributes of the data.

Response 4: D3 is the existing location distance of villages, which is derived from the “village archives” obtained by investigate and survey in 2018. We have marked it in the table.The study was also conducted in 2018.

Point 5:  in Line 135, "referring to the existing research", what's the existing research, which, when, and who?

Response 5: Relevant literature has been supplemented and and marked in section 2.1(11-16).

“11.Zhao, Y.; Zhang, J.; Li, N.; Liang, L. The Study on Conservation Evaluation System and Method on the Historic Cultural Towns&Villages-A Case Study of The First Group Famous Historic and Cultural towns&Villages in China. Acta Geographica Sinica, 2006,26(4),497-505.

12.Zhang, Z.;Yang, Q.; Wang, L.; Su, K.; Kuang, C. Traffic accessibility analysis of traditional villages in minority areas:a case study of Tongren City, Guizhou Province. Resources Science, 2018,40(11),2296-2306.

13.Li, B.; Xu,C.; Zheng,S.; Wang,S.;Dou, Y. Spatial layout characteristics of Ethnic Minority villages based on pattern language: A case study of Southern Dong Area in Xiangxi as an example. Scientia Geographica Sinica,2020,40(11), 1784-1794.

Qin, X.; Li, X.; Chen, W.; Zhang, X. Spatial pattern and influenceing factors of ethnic minority villages in the yangtze river economic belt. Huanman geography, 2022,37(3),118-130.

15.Zhou, T.; Huang, Y.; Wang, X. A analysis of the conservation evaluation system of historic towns and villages in southwest China. Urban Planning Forum, 2011, (6), 109-116.

16.Lei,W. Research on the creation of the features and characteristics of urban waterfront of Xiangxi. Chang Sha: Central South University, 2012.”

Point 6: in line 151, the calculation results of the zoning indexes of each village are obtained by weighting and summing the corresponding indexes of each village with the determined weight of each index.  we cannot know how is the weight calculated.

Response 6: The information is added in Table 2-1 and Table 2-2. For the comprehensive evaluation layer and the factor evaluation layer, the analytic hierarchy process is used to calculate the weight of A, and for the factor evaluation layer, the entropy method is used to calculate the relative weight. The weight of each factor calculated by the two methods is multiplied to obtain the weight of the comprehensive subjective and objective weighting method. For the sake of brevity, the specific calculation process is omitted, refer to related articles[36].

Finally, special thanks to you for your constructive comments again!

Reviewer 2 Report

The paper is well organized, while, I would have to say, it is an interesting topic and a good research topic in rural China, However, the author hasn’t provided a very rigorous research question, research analysis framework and corresponding methodology and it quite like a technical report of the evaluation job, based on these, I can not accept the paper in the current version, my specific comments are in the bellowing:

(1) The abstract hasn’t been well organized, you should simply put forward you core research question, and then your research method and theory used in the paper, and then your research findings and the possible conclusion and policy recommendation for the practice, especially when you research a very typical rural policy in China. In the current status, the abstract is too longer, which need to simplify. In some place the expression, such as “the calculation of the results yields the characteristics of the villages and villages”, “Reference and reference”, need to be re-organize.  

(2) I found a lot of language problems, in the introduction, such as “feature features”, I think the author should be attention for these problems.

(3)  I think when you organized your literature review, it is horrible in the current status of the paper, you cannot list which researches that the authors have done, you should try to make a summarization, regarding on one aspect, a lot of authors have focus on this, and for another aspect, some other authors have focus on this. This is the literature review; it need summarization and then evaluation. Then you can find the research gap, and confirm that the research questions that you focus on are important. Normally, at the last of the paper, we will introduce the arrangement of the context structure, what contents the second section, third section and the last section will present.

(4)  The major problems lie in that the paper has not a very rigorous theoretical framework. I Find that the paper you are talking about is try to divided the villages in Rongcheng into different types and then as for different types of villages, you have put forward corresponding strategies to protect and improve the rural residents and then develop the village, if this is your scientific research problem, You should try to construct a good theoretical framework, which theory you are used, which theory helps you to form a indicators system, and then helps you to make classification for the villages and obey what you put forward some strategies and measure, you should better construct a theoretical framework diagram, which can make the route more clear.

(5) Another major problem lies in that you have not do a mechanism analysis, which is very important for the economic and geographical paper. Yes, you have divided four types of villages, while why these four different villages have formed, the formed mechanism behind the rural houses’ types is very important, which can help you to put forward the targeted measure for the protection and improvement of the rural areas. In the current stage, the contents are lacking.

(6) In the geography picture, you have presented the research areas of your paper, while you haven’t presented the whole picture of China, the south coastal, Taiwan and some other sections are not in your geography picture, this is very serious problem.

(7) I think the paper is quite simple and with a detailed introduce of the rural houses characteristics and then use the indicators system to classified the rural areas, which is not a deep research question that have illustrate the deep influence mechanism, and I am also hold suspect why the different rural areas should take the measures that the paper introduced in that manner.

(8) When do the cluster analysis, normally at the current stage, SPSS 26 has been widely used and the function will be more completed, the author should use the newest analysis tool. In addition, we usually talked about the study areas first and then the methodology, the author needs to adjust the order the context.

(9) Although the authors have talked about some policy tools and possible measure to protect and improve the different villages in the results section, I think the author need to give a clear policy recommendation and listed them in order in the conclusion section, which is need for the government bodies.

(10)           The authors no need to put such a large content to introduce the deficiency and outlook. To my point, the social-economic characters, the housing and residents’ data which are very important in the micro level, so the rural field investigation and questionnaire survey are need with the households and villages. The reference lists are quite less, you should take reference for more excellent and good literature.

Author Response

  1. 2. Replies to reviewer #2

The paper is well organized, while, I would have to say, it is an interesting topic and a good research topic in rural China, However, the author hasn’t provided a very rigorous research question, research analysis framework and corresponding methodology and it quite like a technical report of the evaluation job, based on these, I can not accept the paper in the current version, my specific comments are in the bellowing:

Thank you very much for your good question.

Point 1: The abstract hasn’t been well organized, you should simply put forward you core research question, and then your research method and theory used in the paper, and then your research findings and the possible conclusion and policy recommendation for the practice, especially when you research a very typical rural policy in China. In the current status, the abstract is too longer, which need to simplify. In some place the expression, such as “the calculation of the results yields the characteristics of the villages and villages”, “Reference and reference”, need to be re-organize.  

Response1: We have reorganized the abstract according to your suggestion. Some expressions are condensed, and some important information about data sources, research methods and main conclusions are added and clarified. Modified some inappropriate places and expressions.

Point 2 : I found a lot of language problems, in the introduction, such as “feature features”, I think the author should be attention for these problems.

Response2: We apologize for the poor language of our manuscript. We worked on the manuscript for a long time and the repeated addition and removal of sentence and sections obviously led to poor readability. We have now worked on both language and readability and have also involved native English speakers for language corrections.We really hope that the flow and language level have been substantially improved. 

Point 3:  I think when you organized your literature review, it is horrible in the current status of the paper, you cannot list which researches that the authors have done, you should try to make a summarization, regarding on one aspect, a lot of authors have focus on this, and for another aspect, some other authors have focus on this. This is the literature review; it need summarization and then evaluation. Then you can find the research gap, and confirm that the research questions that you focus on are important. Normally, at the last of the paper, we will introduce the arrangement of the context structure, what contents the second section, third section and the last section will present.

Response3: We have modified the introduction part and sorted out and summarized the existing relevant research, so as to find the deficiencies of the current relevant studies and put forward the research problems of this paper. At the same time, the last paragraph explains the chapter arrangement of the whole article. “The rest of this paper is organized as follows(Fig. 1). Section 2 put forward the concepts and theoretical underpinning, and Section 3 summarizes the sources of data and the selection of typical study area, introduces methodology used in our study. Section 4 measures the characteristic style of rural house in Rongchen, Shandong in 2018. Section 5 discusses the contribution to research, limitation and future work, as well as puts forward policy enlightenment for protection and development of villages with different style. Section 6 draws the conclusions.”

Point 4: The major problems lie in that the paper has not a very rigorous theoretical framework. I Find that the paper you are talking about is try to divided the villages in Rongcheng into different types and then as for different types of villages, you have put forward corresponding strategies to protect and improve the rural residents and then develop the village, if this is your scientific research problem, You should try to construct a good theoretical framework, which theory you are used, which theory helps you to form a indicators system, and then helps you to make classification for the villages and obey what you put forward some strategies and measure, you should better construct a theoretical framework diagram, which can make the route more clear.

Response4: We construct a theoretical basis and research framework, and add the theoretical framework and research route(Part2: Theoretical underpinning). It provides a theoretical basis for the index system construction, classification and strategy proposal.

Figure  Overall flow chart of the method

Point 5:  Another major problem lies in that you have not do a mechanism analysis, which is very important for the economic and geographical paper. Yes, you have divided four types of villages, while why these four different villages have formed, the formed mechanism behind the rural houses’ types is very important, which can help you to put forward the targeted measure for the protection and improvement of the rural areas. In the current stage, the contents are lacking.

Response5: We have enriched and deepened the mechanism analysis in the discussion part, the relationship between characteristic style of rural dwellings and rural revitalization and development, and analyzed the characteristics and formation mechanism of several types of villages.

Figure  Concepts, theoretical framework and interactive mechanism between characteristic style development direction of rural houses and the rural revitalization.

Point 6: In the geography picture, you have presented the research areas of your paper, while you haven’t presented the whole picture of China, the south coastal, Taiwan and some other sections are not in your geography picture, this is very serious problem.

Response6: We have redrawn the map to indicate the relative position of the study area. Rongcheng is located in the easternmost part of Shandong Province and belongs to Weihai. So we marked its location in Shandong Province.

Figure  Study area

Point 7:  I think the paper is quite simple and with a detailed introduce of the rural houses characteristics and then use the indicators system to classified the rural areas, which is not a deep research question that have illustrate the deep influence mechanism, and I am also hold suspect why the different rural areas should take the measures that the paper introduced in that manner.

Response7: We have raised the research question in part 1. “The purpose of this paper is to study the characteristic style of rural houses in coastal city. This paper attempts to construct an index system suitable for the characteristic style of rural dwellings, takes the Rongcheng City, Shandong as the research area, evaluates the rural houses’ characteristics of 17 villages integrating the  multidimensional data,including statistical data and survey measured data in 2018, .divides the style types of village dwellings, reveals the characteristics and development direction of each region, provides policy recommendations for subsequent targeted protection and development,enriches the research in the style field of rural houses’ characteristic. Through quantitative evaluation and zoning research, the development conditions and directions are clarified, and corresponding measures are set to protect and enhance the value of the rural characteristic style, which is connected with the rural revitalization strategy and complements the construction of beautiful China”.

The construction of evaluation and zoning index system reflects the characteristics of each village and the forming environment of the characteristic style of folk houses.

Point 8 : When do the cluster analysis, normally at the current stage, SPSS 26 has been widely used and the function will be more completed, the author should use the newest analysis tool. In addition, we usually talked about the study areas first and then the methodology, the author needs to adjust the order the context.

Response8: Since it is the most basic function of the software, we previously considered that it can complete the goal, but did not consider the old and new problems of the version. We re-did the cluster analysis with the latest version.

We adjusted the order the context, which is the study areas first and then the methodology.

Point 9: Although the authors have talked about some policy tools and possible measure to protect and improve the different villages in the results section, I think the author need to give a clear policy recommendation and listed them in order in the conclusion section, which is need for the government bodies.

Response9: We have raised the research question in part 1. We have revised the results and discussion sections and extracted clear and concise policy recommendations, and drawn the mechanism diagram, which are summarized in the conclusion. It is expected to be helpful to theory and practice on characteristic style of coastal rural house.

“The development of rural areas is an important foundation for China 's economic development. The protection and development management of residential features in traditional villages is an important part of rural development. The characteristic features of rural dwellings embody the historical and cultural value of the village. Through quantitative evaluation and zoning study, the development conditions and directions are defined, and corresponding measures are set up to protect and enhance the value of the features. On the one hand, this can be connected with the rural revitalization strategy, and it is more complementary to the construction of beautiful China ; on the other hand, this is also the specific path to implement the ' Weihai Beautiful Village Planning and Design Technical Guidelines ' and ' Rongcheng Urban Rural Construction Plan (2017-2030) . This will not only promote the protection and construction of the residential features of coastal traditional villages, but also provide reference directions and measures for the employment of village industries and farmers, the long-term stable increase of local people 's income, and living and working in peace and contentment.”

Point 10 : The authors no need to put such a large content to introduce the deficiency and outlook. To my point, the social-economic characters, the housing and residents’ data which are very important in the micro level, so the rural field investigation and questionnaire survey are need with the households and villages. The reference lists are quite less, you should take reference for more excellent and good literature.

Response10: We rearranged the content, simplifying the research deficiencies and prospects. More emphasis on the analysis of socio-economic, housing and resident characteristics; This paper analyzes the understanding and cognition of villages and residents collected and felt during the village survey in the process of the project. This area is a typical distribution area of seagrass house. The origin and characteristics of seagrass house are briefly introduced. Drawing the real scene and the typical picture of the painted sea grass house is helpful to perceptual cognition.

Finally, special thanks to you for your constructive comments again!

Reviewer 3 Report

         This is an interesting and meaningful study. In the context of rural revitalization, we should pay attention to the protection of coastal villages. Here are some suggestions for your reference:

         (1) The introduction needs a modest rewrite. First, compared with the existing research, the marginal contribution of this study is not clear, so the author is suggested to further condensing. The marginal contribution can be from the theoretical perspective, the research content or the research method. Second, it is not clear what the key scientific questions to be addressed in this study are and need to be further clarified.

         (2) The theoretical analysis framework needs further improvement. At present, the author's selection of indicators covers various data, which data is completed under the guidance of a large theoretical analysis framework needs further introduction, otherwise, the selection and construction of the whole index system will appear to lack of theoretical support.

         (3) The introduction of data sources needs to be more detailed. The authors use a variety of data, the sources of which need to be described in more detail, especially the survey data. How to ensure that the selected sample can be good overall characteristics, can be repeated, this is the author needs to consider, but also the reader's concern.

Author Response

  1. 3. Replies to reviewer #3

This is an interesting and meaningful study. In the context of rural revitalization, we should pay attention to the protection of coastal villages. Here are some suggestions for your reference:

         Point 1: The introduction needs a modest rewrite. First, compared with the existing research, the marginal contribution of this study is not clear, so the author is suggested to further condensing. The marginal contribution can be from the theoretical perspective, the research content or the research method. Second, it is not clear what the key scientific questions to be addressed in this study are and need to be further clarified.

Response 1: Thanks for your careful work.

We rewrote the introduction, combed and summarized the existing research, found the research deficiencies, and put forward the research questions of this paper. Thus, contributions to the theoretical and practical aspects of this study can be obtained.

“The purpose of this paper is to study the characteristic style of rural houses in coastal city. This paper attempts to construct an index system suitable for the characteristic style of rural dwellings, takes the Rongcheng City, Shandong as the research area, evaluates the rural houses’ characteristics of 17 villages integrating the  multidimensional data,including statistical data and survey measured data in 2018, .divides the style types of village dwellings, reveals the characteristics and development direction of each region, provides policy recommendations for subsequent targeted protection and development,enriches the research in the style field of rural houses’ characteristic. Through quantitative evaluation and zoning research, the development conditions and directions are clarified, and corresponding measures are set to protect and enhance the value of the rural characteristic style, which is connected with the rural revitalization strategy and complements the construction of beautiful China”.

Point 2: The theoretical analysis framework needs further improvement. At present, the author's selection of indicators covers various data, which data is completed under the guidance of a large theoretical analysis framework needs further introduction, otherwise, the selection and construction of the whole index system will appear to lack of theoretical support.

Response 2: It adds the theoretical analysis frame and provides the basis for the construction of index system. Also further explains the writing motivation and logic of the article.(Part 2)

Figure  Overall flow chart of the method.

         Point3: The introduction of data sources needs to be more detailed. The authors use a variety of data, the sources of which need to be described in more detail, especially the survey data. How to ensure that the selected sample can be good overall characteristics, can be repeated, this is the author needs to consider, but also the reader's concern.

Response 3: The paper explains the source of data, the typicality of the study area and the representativeness of villages and dwellings.

“The three-dimensional laser data is the real data of the scanned seagrass house. In the later stage, the data is processed and modeled to achieve the effect of restoring the real scene. 3D laser scanning is the first step to build the model, and the scale parameters of the buildings obtained need to be reconstructed using specialized modeling software, which is the second step of modeling. The modeling software used in this paper is SKETCHUP, the Chinese name is Sketchup Master. It has a wide user base all over the world and can easily and quickly build house models. The last step in building the seagrass house model is the texture mapping of the seagrass house model surface.

Auxiliary analysis data including DEM data, population and socio-economic data, land use data and various types of planning data and so on. The DEM data are downloaded from the website (http://www.resdc.cn), population, GDP and other socio-economic data are mainly obtained from the statistical yearbook and field survey of Rongcheng City ( Table 2-1). It is used to analyze the economic development status and housing bearing capacity of characteristic villages in Rongcheng City, and it has an important reference for the evaluation of characteristic buildings. The planning data show the development direction of the region and relevant policies issued by the state, etc., which can help this paper evaluate and regionalize the characteristic features of the coastal region scientifically and reasonably. The planning data used in this paper mainly include: general planning of land use in Rongcheng City, rural construction planning in Rongcheng City, etc. All kinds of measured data, planning data, population social and economic data are statistically analyzed to provide basic data for quantitative evaluation of characteristic features.”

Finally, special thanks to you for your constructive comments again!

Reviewer 4 Report

Manuscript ID: ijerph-2168777

Title:Evaluation of rural houses’ characteristics in coastal villages and its inspiration to rural revitalization: Case study of Rongcheng in Shandong Province

Comments:

The characteristics of rural houses are an important manifestation of the historical and cultural values of rural areas, and are the key focus of the implementation of the strategy for the construction of beautiful China and the revitalization of rural areas. The coastal rural villages in Rongcheng, represented by seaweed houses, have become an important feature of this city, and its research has received increasing attention.

This paper represents a good attempt to study the evaluates the rural houses' characteristics in coastal villages based on the existing data and research foundations, which a suitable evaluation index system for the region has been constructed. The choice of quantitative methodology permits interesting insights to be drawn, that is, The overall evaluation of the village's overall environment, coastal construction value, and traditional folk culture has been used as a comprehensive evaluation factor.

The paper is well organized, it has an innovative topic, clear and reasonable analytical framework, showing the strength and value of case studies at county scales, which is basically relevant to the theme of this journal.

Some minor suggestions for the further improvement of the manuscript. The article is recommended to be accepted after revision.

(1)  Abstract:

The first three sentences which are the background, significance and process of the research should be refined and reduced appropriately. The research results need to be further refined and clearly expressed.

(2)  Introduction:

 In general, the article presents with a relatively comprehensive bibliography. It is still suggested that the author supplement the latest representative literature on the characteristics of rural dwellings and its relationship with rural revitalization, which can improve the literature review.

(3)  Methods, Region and Data Processing:

In the section of methods, the expression of index system on evaluation and zoning of rural characteristic dwellings need to be further distinguished and streamlined.

The section of study area is too much,which can be summarized. It is suggested that focus on the rural characteristic dwellings and rural development.

Figure2-1 needs to be redrawn to make it more standard, beautiful and informative.

(4)  Result and analysis, Discussion and Implications

Relevant statements need to be simplified and merged to make the expressions coherent. The words and expressions of key nouns should be unified before and after the full text, such as “rural houses’ characteristics in coastal villages”, however the word “feature” is used in some places in the text instead of characteristics. Enhancing the cohesion and correspondence between the two parts: 3.2 Coastal rural residential features division and 4 Construction Direction of Different Landscape Areas.

In the section of discussion, a mechanism diagram should be added to express the relationship between the evaluation of rural characteristics and rural revitalization. At the same time, the text should also correspond to the relevant elaboration and discussion.

The figure 3-1 and figure 4-1 should be redrawn or reselected, in order to better express the meaning of its carrying.

There is short of reference of research results to similar research topics carried out in other parts of the world, the research is local but the problem is global, and comparisons should be extended to other regions of the globe, which can further show how these studies differ from other studies already done, what is innovative in them.

(5)  I think in its present form, the manuscript needs minor English grammar corrections and several additions, the authors need to find a native speaker to check the language. At the same time, the expression of full text should be consistent on the main concepts.

Author Response

  1. 4. Replies to reviewer #4

The characteristics of rural houses are an important manifestation of the historical and cultural values of rural areas, and are the key focus of the implementation of the strategy for the construction of beautiful China and the revitalization of rural areas. The coastal rural villages in Rongcheng, represented by seaweed houses, have become an important feature of this city, and its research has received increasing attention.

This paper represents a good attempt to study the evaluates the rural houses' characteristics in coastal villages based on the existing data and research foundations, which a suitable evaluation index system for the region has been constructed. The choice of quantitative methodology permits interesting insights to be drawn, that is, The overall evaluation of the village's overall environment, coastal construction value, and traditional folk culture has been used as a comprehensive evaluation factor.

The paper is well organized, it has an innovative topic, clear and reasonable analytical framework, showing the strength and value of case studies at county scales, which is basically relevant to the theme of this journal.

Some minor suggestions for the further improvement of the manuscript. The article is recommended to be accepted after revision.

Point 1: Abstract:

The first three sentences which are the background, significance and process of the research should be refined and reduced appropriately. The research results need to be further refined and clearly expressed.

 Response 1: Thanks for your recommendation and affirmation.

We have streamlined the relevant parts. The results were streamlined and reorganized.

We simply put forward core research question, and then research method and theory used in the paper, and then research findings and the possible conclusion and policy recommendation for the practice.

Point 2: Introduction:

 In general, the article presents with a relatively comprehensive bibliography. It is still suggested that the author supplement the latest representative literature on the characteristics of rural dwellings and its relationship with rural revitalization, which can improve the literature review.

Response 2: We have supplemented some of the literature on the relationship between rural settlements and rural revitalization.

It adds the theoretical analysis frame and provides the basis for the construction of index system. Also further explains the writing motivation and logic of the article.(Part 2)

Figure Overall flow chart of the method.

Point 3: Methods, Region and Data Processing:

In the section of methods, the expression of index system on evaluation and zoning of rural characteristic dwellings need to be further distinguished and streamlined.

The section of study area is too much,which can be summarized. It is suggested that focus on the rural characteristic dwellings and rural development.

Figure2-1 needs to be redrawn to make it more standard, beautiful and informative.

Response 3: We have added the theoretical basis of index system construction. The indexes of evaluation and regionalization are explained respectively(part 2).

“The characteristic style of rural residential houses is formed based on certain geographical space, combined with characteristic historical and cultural factors. The overall village environment is a comprehensive expression of rural location, village pattern, streets and courtyards, and reflects the pattern and form of the village. The characteristic style value of rural dwellings is the sum of tangible and intangible values of the whole village, which is mainly reflected by architectural value and traditional cultural value. The main style of coastal residential in Rongcheng is that there are a large number of sea grass houses. The distribution and preservation status of sea grass houses in different villages are different, so each has its own characteristics. Therefore, this study needs to take into account the characteristics of coastal buildings in the quantitative style evaluation of coastal rural residential, while, combine with the characteristics of folk culture to build a suitable evaluation index system. Evaluation and zoning is to provide spatial guidance for the implementation of differentiated management policies, so it is necessary to consider the protection, development and management of rural residential features, so as to make it more targeted and practical.”

The study area is simplified, focusing on rural features and village development profiles.

The location map of study area 2-1 was redrawn.

Point 4:  Result and analysis, Discussion and Implications

Relevant statements need to be simplified and merged to make the expressions coherent. The words and expressions of key nouns should be unified before and after the full text, such as “rural houses’ characteristics in coastal villages”, however the word “feature” is used in some places in the text instead of characteristics. Enhancing the cohesion and correspondence between the two parts: 3.2 Coastal rural residential features division and 4 Construction Direction of Different Landscape Areas.

In the section of discussion, a mechanism diagram should be added to express the relationship between the evaluation of rural characteristics and rural revitalization. At the same time, the text should also correspond to the relevant elaboration and discussion.

The figure 3-1 and figure 4-1 should be redrawn or reselected, in order to better express the meaning of its carrying.

There is short of reference of research results to similar research topics carried out in other parts of the world, the research is local but the problem is global, and comparisons should be extended to other regions of the globe, which can further show how these studies differ from other studies already done, what is innovative in them.

Response4: We have modified the expression of the relevant part. The relationship between parts 3.2 and 4 is corresponding before and after. In the four parts, we re-analyzed the mechanism and drew the mechanism diagram. 3-1 and 4-1 have been reselected and drawn, making it more accurately. In the introduction part, relevant domestic and foreign references and theoretical framework are added, and comparative analysis is carried out in the discussion, which makes the article more universal and international.

Figure  Concepts, theoretical framework and interactive mechanism between characteristic style development direction of rural houses and the rural revitalization.

Point 5:  I think in its present form, the manuscript needs minor English grammar corrections and several additions, the authors need to find a native speaker to check the language. At the same time, the expression of full text should be consistent on the main concepts.

Response 5: We apologize for the poor language of our manuscript. We worked on the manuscript for a long time and the repeated addition and removal of sentence and sections obviously led to poor readability. We have now worked on both language and readability and have also involved native English speakers for language corrections.We really hope that the flow and language level have been substantially improved. 

Finally, special thanks to you for your constructive comments again!

Round 2

Reviewer 2 Report

The paper is well organized, while, I can find that the authors have done great efforts for the modification and efforts for the paper, while, to be a high quality research article, I keep some major concerns for the manuscript:

(1) What is the main research meaning for divide the rural houses into different types according to the three major factors, does it show great importance for the scientific theory research or for the practical policy recommendation? In fact, the local government may have developed different regulations and plans according to the different characteristics in different areas. Dose the research can provide some suggestions for the revisions of the governments’ development plan?   

(2) In my opinions, the AHP and entropy methods are very ordinary and easy methods to construct an index system and give the weights of the different levels of index. That is to say, the methodology of the paper is not sufficient and very simple, and the paper only do an empirical job: to divide the rural house into different regions types. By using a simple method to accomplish a classification work, which lacks of the wide interest and attraction for the readers and academics.

(3) Once again: the major problem lies in that you have not do a mechanism analysis, which is very important for the economic and geographical paper. the formed mechanism behind the region types of the rural houses is very important, which factors have resulted in the formation of this type in these areas, which can include the regional economic factors, nature, human, society and also other factors, you can make a regression analysis to find out the major influencing factors, which can help you to give some policy recommendation in different areas and different regional types.

(4) The authors have constructed an overall flow chart of the method, while it is not the theoretical framework of the whole paper, and it cannot represent the conceptual framework, which is very important for a scientific paper, in this framework, you should mention the theory you used, the theoretical bases, the formation routes, and influence effects, ……. That is to say, why the factors drive the rural areas into the specific region types? In the geography picture, you have presented the research areas of your paper, However, you should put the whole picture of China in the right-down corner of the picture.

(5) Figure 7 have included a lot of information, while it cannot well integrate the topic of this paper. How can we put the theoretical framework in the result sections? And how does the contents link to rural vitalization, we haven’t talked about this in the introduction and research question, Normally, the research limitations and research prospects should put after the conclusion and policy recommendations, For such a practical applied articles, the authors should put a lot of sentences for the policy and planning suggestions for the different regional types with different rural house characteristics, these are very important after your classification for the rural houses areas and regional types.

Author Response

Manuscript ID: ijerph-2168777

Title: Evaluation of rural houses’ characteristics in coastal villages and its inspiration to rural revitalization: Case study of Rongcheng in Shandong Province

Journal: International Journal of Environmental Research and Public Health;

Section “Environmental Science and Engineering”; 

Special Issue “Land Use Change and Its Effects on Regional Sustainable Development”.

List of Changes

First of all, we appreciate your careful review and positive comments! All of your comments are very important and really helpful to revise and improve our manuscript (MS). According to your comments, we have made relevant changes (words in red) to the whole MS.The detailed revisions are listed below, responding to your comments point by point.

  1. 2. Replies to reviewer #2

The paper is well organized, while, I can find that the authors have done great efforts for the modification and efforts for the paper, while, to be a high quality research article, I keep some major concerns for the manuscript:

Thank you very much for your constructive comments.

Point 1:  What is the main research meaning for divide the rural houses into different types according to the three major factors, does it show great importance for the scientific theory research or for the practical policy recommendation? In fact, the local government may have developed different regulations and plans according to the different characteristics in different areas. Dose the research can provide some suggestions for the revisions of the governments’ development plan?

Response1: We have elaborated frame construction diagram of indicator system of rural houses’ characteristic style(Figure 2.). The characteristic style of rural houses is formed under certain geographical space, combined with characteristic historical and cultural factors. “Evaluation index system of Chinese historical and cultural famous towns (famous villages)” and “Evaluation and Recognition Index System of Traditional Villages (Trial)” issued by Ministry of Housing and Urban-Rural Development of the People's Republic of China, where construct the index system from material and intangible cultural heritage to evaluatevalue characteristic of the village(town). Literature [11-16] also provides us with important references.

The regionalization of characteristic features of rural house should first consider the value of characteristic style of rural house, then combine the status quo of protection, development and management, further analyze the socio-economic conditions and location conditions of villages, and finally consider the overall environment and natural conditions of villages, so as to comprehensively identify different villages and divide type areas. The regionalization is based on 6 comprehensive evaluation layer Bn.

Regionalization is to provide spatial guidance for the implementation of differentiated management policies, so it is necessary to consider the protection, development and management of rural house characteristic style, so as to make regionalization more targeted and practical. this can be connected with the rural revitalization strategy, and it is more complementary to the construction of beautiful China; on the other hand, this is also the specific path to implement the ' Weihai Beautiful Village Planning and Design Technical Guidelines ' and ' Rongcheng Urban Rural Construction Plan (2017-2030) . This will not only promote the protection and construction of the residential features of coastal traditional villages, but also provide reference directions and measures for the employment of village industries and farmers, the long-term stable increase of local people 's income, and living and working in peace and contentment.

Point 2: In my opinions, the AHP and entropy methods are very ordinary and easy methods to construct an index system and give the weights of the different levels of index. That is to say, the methodology of the paper is not sufficient and very simple, and the paper only do an empirical job: to divide the rural house into different regions types. By using a simple method to accomplish a classification work, which lacks of the wide interest and attraction for the readers and academics.

Response2: You're right. However, these two methods can meet our analysis needs. We combine the two methods, quantitative and qualitative to determine the weight. The index weight determined by subjective weighting method can reflect the intention of researchers, but human factors limit the application of actual data. However, the weights determined by objective weighting method are closely related to actual data, but are susceptible to extreme values[36,38]. Therefore, this study uses a comprehensive weight determination method combining subjective and objective weighting methods, and chooses the classical subjective weighting method--analytic hierarchy process (AHP) and the objective weighting method reflecting actual data--entropy method. Combining the two methods can complement each other and reduce the limitations of a single method. The index system constructed in this study consists of four layers. For the comprehensive evaluation layer(B) and the factor evaluation layer(C), the analytic hierarchy process is used to calculate the weight to target layer(A). For the sub factor evaluation layer(D), the entropy method is used to calculate the relative weight. The weight of each factor calculated by the two methods is multiplied to obtain the weight of the comprehensive subjective and objective weighting method.

Point 3: Once again: the major problem lies in that you have not do a mechanism analysis, which is very important for the economic and geographical paper. the formed mechanism behind the region types of the rural houses is very important, which factors have resulted in the formation of this type in these areas, which can include the regional economic factors, nature, human, society and also other factors, you can make a regression analysis to find out the major influencing factors, which can help you to give some policy recommendation in different areas and different regional types.

Response3: We have further perfect mechanism analysis. The index selection basis is added, the situation of each regional village is supplemented, and the interactive relationship with rural revitalization is analyzed.

Part 2.2  “Evaluation index system of Chinese historical and cultural famous towns (famous villages)” and “Evaluation and Recognition Index System of Traditional Villages (Trial)” issued by Ministry of Housing and Urban-Rural Development of the People's Republic of China, where construct the index system from material and intangible cultural heritage to evaluatevalue characteristic of the village(town). Literature [11-16] also provides us with important references.

Figure 2. Frame construction diagram of indicator system of rural houses’ characteristic style.

Part 3.3.1  The characteristic style of coastal rural house are evaluated from three aspects: the overall village environment, the value of coastal building and traditional folk culture. The overall village environment reflects the pattern and morphological structure of the village, including environmental style and overall pattern, and the spatial form of streets and courtyards. The former selects three indices: the beauty of the village natural environment, the preservation degree of the traditional village pattern, and the longevity of the existing village, while the latter selects two indices: the total length of traditional streets, the total area of traditional courtyards. The coastal buildings value includes two parts: historical value and characteristic value. The former is evaluated by the grade of cultural relic protection unit and the earliest building age of seagrass houses, while the latter is evaluated by three indices: The area percentage of seagrass houses to village buildings, the quantity percentage of sea grass houses and percentage of seagrass house in good quality. Traditional folk culture includes three parts: the uniqueness of traditional folk culture, the maintenance degree of traditional folk skills, and the persistence degree of traditional life mode. The quantity and permanence of customs, festivals, legends, and poems, the quantity and permanence of handicrafts, traditional arts and performance forms, and the proportion of the population living in seagrass houses to the permanent population and the proportion of the population engaged in traditional production methods such as agriculture and fishery are selected respectively.

Part 3.3.2  Under the research of relevant scholars and the advice of experts[11,12], the regionalization of characteristic features of rural house should first consider the value of characteristic style of rural house, then combine the status quo of protection, development and management, further analyze the socio-economic conditions and location conditions of villages, and finally consider the overall environment and natural conditions of villages, so as to comprehensively identify different villages and divide type areas.

Figure 8. Interaction mechanism between characteristic style development direction of rural houses and the rural revitalization.

Part 5.1  “The protection of traditional villages allows us to retain the villagers, protect the countryside and remember homesickness. Through traditional villages, an evolving framework of rural revitalization and cultural routes can be established, in terms of the inherent properties of both(Figure 8.). First, among the four major types of rural revitalization promotion, the feature conservation type basically refers to traditional villages with historical value and cultural heritage. Second, traditional villages constitute the elements of cultural routes from points to line. These villages as elements must share the associated characteristics of cultural routes in three ways. Finally, cultural routes have striking potential to promote rural revitalization from line to region, through an evolving strategy[41]. As an important part of the rural regional system, traditional villages are the main positions of rural revitalization. The transformation and development of their living environment is a stress response under the national strategic blueprint such as beautiful villages and new urbanization, and the only way to comply with the requirements of The Times and the objective laws of villages[42]. ”

Point 4: The authors have constructed an overall flow chart of the method, while it is not the theoretical framework of the whole paper, and it cannot represent the conceptual framework, which is very important for a scientific paper, in this framework, you should mention the theory you used, the theoretical bases, the formation routes, and influence effects, ……. That is to say, why the factors drive the rural areas into the specific region types? In the geography picture, you have presented the research areas of your paper, However, you should put the whole picture of China in the right-down corner of the picture.

Response4: We construct a theoretical basis and research framework, and add the theoretical framework and research route(Part2: Theoretical underpinning; Part5 Interaction mechanism between characteristic style development direction of rural houses and the rural revitalization.). It provides a theoretical basis for the index system construction, classification and strategy proposal.

We have put the whole picture of China in the right-up corner of the picture.

Figure 2. Frame construction diagram of indicator system of rural houses’ characteristic style.

Figure 3. Study area

Figure 8. Interaction mechanism between characteristic style development direction of rural houses and the rural revitalization.

Point 5:  Figure 7 have included a lot of information, while it cannot well integrate the topic of this paper. How can we put the theoretical framework in the result sections? And how does the contents link to rural vitalization, we haven’t talked about this in the introduction and research question, Normally, the research limitations and research prospects should put after the conclusion and policy recommendations, For such a practical applied articles, the authors should put a lot of sentences for the policy and planning suggestions for the different regional types with different rural house characteristics, these are very important after your classification for the rural houses areas and regional types.

Response5: We have condensed and deepened the mechanism analysis in the discussion part,  interaction mechanism between characteristic style development direction of rural houses and the rural revitalization., and analyzed the characteristics and formation mechanism of several types of villages.

We have put a lot of sentences for the policy and planning suggestions for the different regional types with different rural house characteristics, put the research limitations and research prospects after the conclusion and policy recommendations as 6.2. The policy Recommendations section is supplemented as 5.2.

Part 5.1  “Through traditional villages, an evolving framework of rural revitalization and cultural routes can be established, in terms of the inherent properties of both(Figure 8.). First, among the four major types of rural revitalization promotion, the feature conservation type basically refers to traditional villages with historical value and cultural heritage. Second, traditional villages constitute the elements of cultural routes from points to line. These villages as elements must share the associated characteristics of cultural routes in three ways. Finally, cultural routes have striking potential to promote rural revitalization from line to region, through an evolving strategy[41]. As an important part of the rural regional system, traditional villages are the main positions of rural revitalization. The transformation and development of their living environment is a stress response under the national strategic blueprint such as beautiful villages and new urbanization, and the only way to comply with the requirements of The Times and the objective laws of villages[42]. ”

Part 5.2  “Each region should be combined with the existing rural construction foundation, the style shape highlights the 'characteristic', that is, to highlight the traditional characteristics, create regional characteristics, heritage national characteristics, highlight the cultural characteristics of organic unity, adjust measures to local conditions[39]. Relying on the rich landscape resources of mountains and rivers, the rural characteristics with obvious landscape division and remarkable landscape characteristics are constructed. (1) Historical culture+industrial development characteristic area. This region......; (2) Folk customs+industrial development characteristic area. This area includes.....; (3) Natural scenery characteristic area. The region is located in.....; (4) Folk customs characteristic area. This area includes Quge Village......”

“The development of rural areas is an important foundation for rural revitalization and even China 's economic development. The protection and development management of residential features in traditional villages is an important part of rural development. The characteristic style of rural house embody the historical and cultural value of the village. Through quantitative evaluation and regionalization study, the development conditions and directions are defined, and corresponding measures are set up to protect and enhance the value of the style. On the one hand, this can be connected with the rural revitalization strategy, and it is more complementary to the construction of beautiful China; on the other hand, this is also the specific path to implement the ' Weihai Beautiful Village Planning and Design Technical Guidelines ' and ' Rongcheng Urban Rural Construction Plan (2017-2030) . This will not only promote the protection and construction of the residential features of coastal traditional villages, but also provide reference directions and measures for the employment of village industries and farmers, the long-term stable increase of local people 's income, and living and working in peace and contentment. ”

Figure 8. Interaction mechanism between characteristic style development direction of rural houses and the rural revitalization.

Finally, special thanks to you for your constructive comments again!

Reviewer 3 Report

I have no other comments, thank you.

Author Response

List of Changes

First of all, we appreciate your careful review and positive comments! All of your comments are very important and really helpful to revise and improve our manuscript (MS). According to your comments, we have made relevant changes (words in red) to the whole MS.The detailed revisions are listed below, responding to your comments point by point.

  1. 2. Replies to reviewer #3

Point 1: 

Respinse1:  We worked on the manuscript for a long time and the repeated addition and removal of sentence and sections obviously led to poor readability.We have now worked on both language and readability and have also involved native English speakers for language corrections.We really hope that the flow and language level have been substantially improved.

Special thanks to you for your constructive and positive comments again!
